# Homeodomain protein Six4 prevents the generation of supernumerary *Drosophila* type II neuroblasts and premature differentiation of intermediate neural progenitors

**Rui Chen, Yanjun Hou** [¤], **Marisa Connell, Sijun Zhu** *

Department of Neuroscience and Physiology, State University of New York Upstate Medical University, Syracuse, New York, United States of America

¤ Current address: Jiangsu Key Laboratory for Molecular and Medical Biotechnology, College of Life Sciences, Nanjing Normal University, Nanjing, China
* zhus@upstate.edu

**Data Availability Statement:** All relevant data are within the manuscript and its Supporting Information files.

## Abstract

In order to boost the number and diversity of neurons generated from neural stem cells, intermediate neural progenitors (INPs) need to maintain their homeostasis by avoiding both dedifferentiation and premature differentiation. Elucidating how INPs maintain homeostasis is critical for understanding the generation of brain complexity and various neurological diseases resulting from defects in INP development. Here we report that Six4 expressed in *Drosophila* type II neuroblast (NB) lineages prevents the generation of supernumerary type II NBs and premature differentiation of INPs. We show that loss of Six4 leads to supernumerary type II NBs likely due to dedifferentiation of immature INPs (imINPs). We provide data to further demonstrate that Six4 inhibits the expression and activity of PntP1 in imINPs in part by forming a trimeric complex with Earmuff and PntP1. Furthermore, knockdown of Six4 exacerbates the loss of INPs resulting from the loss of PntP1 by enhancing ectopic Prospero expression in imINPs, suggesting that Six4 is also required for preventing premature differentiation of INPs. Taken together, our work identified a novel transcription factor that likely plays important roles in maintaining INP homeostasis.

## Author summary

Intermediate neural progenitors (INPs) are descendants of neural stem cells that can proliferate for a short term to amplify the number of nerve cells generated in the brain. INPs play critical roles in determining how big and complex a brain can grow. To perform their function, INPs need to maintain their own population and must not adopt the identity of neural stem cells, a process called dedifferentiation, or acquire the fate of their own daughter cells and stop proliferation too soon, a process called premature differentiation. However, how INPs avoid dedifferentiation and premature differentiation is not fully understood. In this study, we identified a protein called Six4 as a novel factor that plays important roles in preventing the generation of extra neural stem cells and premature

**Funding:** This work was supported by the National Institute of Neurological Disorders and Stroke of the National Institutes of Health under Award Numbers R01NS085232 (S.Z.) and R21NS109748 (S.Z.) and startup funds from SUNY Upstate Medical University (to SZ). The funders did not play any roles in the design, data collection and analysis, decision to publish, or preparation of the manuscript of this study.

**Competing interests:** The authors have declared that no competing interests exist.

differentiation of INPs in developing fruit fly brains. We described how Six4 functionally and physically interacts with other factors that are involved in regulating INP cell fate specification. Our work provides novel insights into the mechanisms regulating INP development and could have important implications in understanding how complex brains are generated during normal development and how abnormal brain development or brain tumor can occur when INPs fail to avoid premature differentiation or dedifferentiation.

## Introduction

Intermediate neural progenitor (INP) cells are fate-restricted progenitor cells generated from neural stem cells (NSCs). INPs serve to amplify the diversity and number of neurons generated from NSCs through transit proliferation, thus contributing to the generation of brain complexity [1,2]. To perform their function, INPs must overcome two challenges. One is to avoid dedifferentiation and reversion back to the NSC fate. Another is to avoid premature differentiation and cell cycle exit. Proper balance of self-renewal, differentiation, and dedifferentiation is crucial for maintaining the homeostasis and population of INPs. Elucidating the cellular and molecular mechanisms regulating the self-renewal, differentiation, and dedifferentiation of INPs will not only provide insights into the generation of brain complexity, but also help us understand cellular and molecular mechanisms of various neurodevelopmental disorders, such as microcephaly and other cortical malformations, as well as the development of brain tumors, which could originate from dedifferentiation of intermediate progenitors [3–6].

In the developing *Drosophila* larval brains, a particular type of NSCs called type II neuroblasts (NBs) produce neurons by generating transit amplifying INPs like NSCs in mammalian brains. These INPs can divide up to 10 rounds in a self-renewing manner to produce terminally dividing ganglion mother cells (GMCs), which further divide once to produce neurons or glia [7–9]. By generating INPs, type II NBs produce much more diverse types of neurons [10–12] than type I NBs, which produce neurons by generating GMCs directly and can be distinguished from type II NBs by their expression of proneural protein Asense (Ase) (Fig 1A) [13]. Thus, *Drosophila* INPs have a similar function as mammalian INPs in amplifying neuronal output from NSCs. Since they were discovered over a decade ago, *Drosophila* type II NB lineages have become an excellent model system for studying INP development.

In order to amplify the neuronal output from type II NBs, INPs in *Drosophila* larval brains also need to avoid dedifferentiation and premature differentiation. INPs that are newly generated from type II NBs are immature and need to undergo a differentiation process to first turn on the expression of Ase and then the self-renewing factors such as bHLH family transcription factors Deadpan (Dpn) and the Notch downstream targets E(spl) family proteins. The immature INPs (imINPs) are genetically unstable and prone to dedifferentiate back to the type II NB fate before they become fate-committed mature INPs (mINPs). Once they become mINPs, they have to maintain self-renewal and avoid exiting the cell cycle prematurely before enough neurons or glial cells are generated. To avoid dedifferentiation, imINPs need to turn off the expression of self-renewing factors such as Dpn and E(spl) proteins right after they are born [14–17]. The elimination of Dpn expression in imINPs involves Brain tumor (Brat)-mediated degradation of dpn mRNAs and degradation of Dpn protein by the ubiquitin-proteosome system [18,19], whereas the suppression of E(spl) expression involves termination of Notch signaling by Numb-mediated degradation of Notch receptor [9,20] and prevention of aberrant Notch activation by retromer complex-mediated trafficking of Notch [21]. Elimination of Dpn

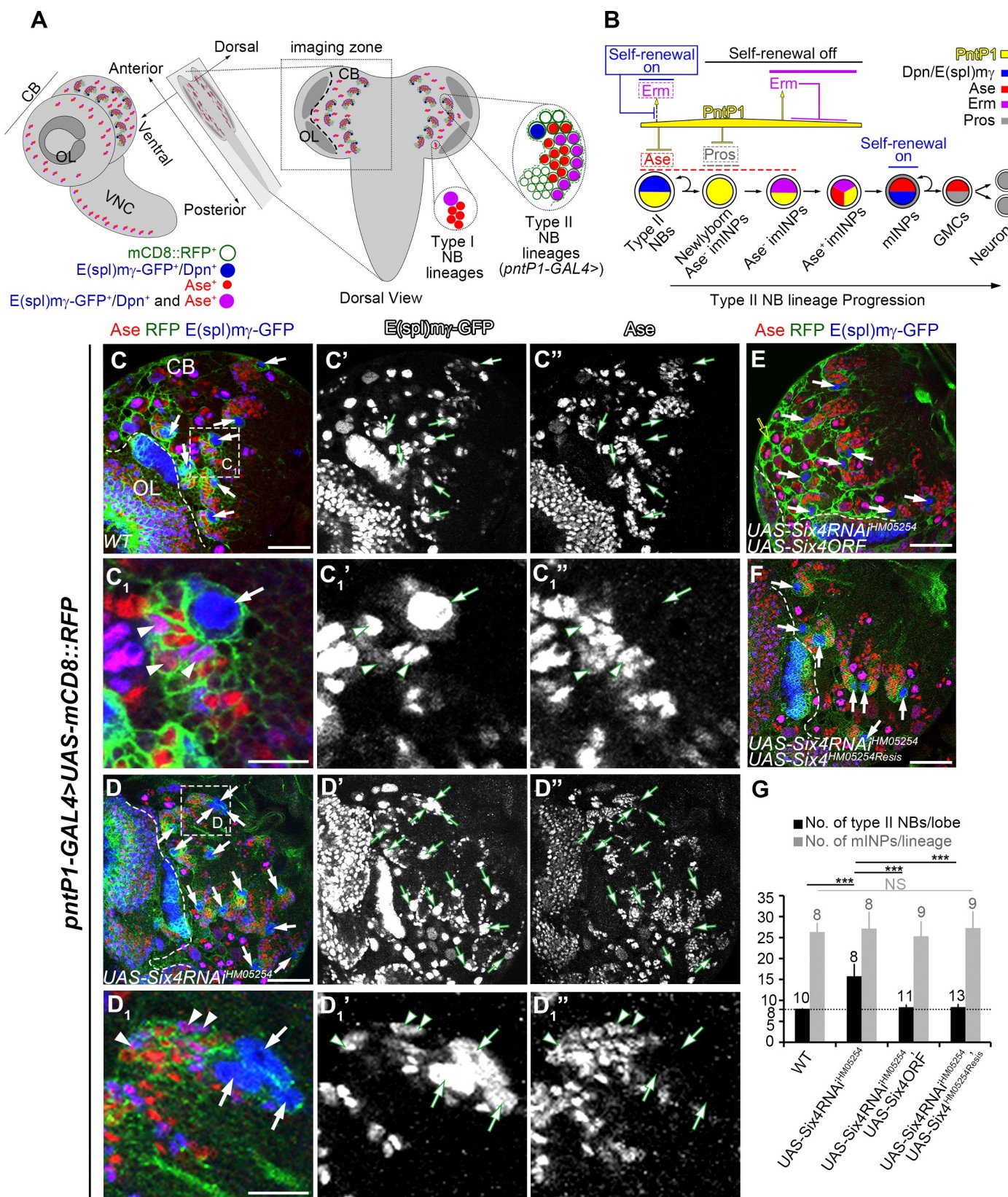

**Fig 1. Knockdown of Six4 results in supernumerary type II NBs.** In all images, type II NB lineages were labeled with mCD8-RFP (in green) driven by *pntP1-GAL4* and counterstained with anti-GFP (in blue) and anti-Ase (in red) antibodies. White arrows point to type II NBs that express E(spl)mγ-GFP but not Ase and appear blue in the nucleus. Type I NBs and mINPs express both Dpn and Ase and appear magenta in the nucleus, whereas GMCs express Ase but not Dpn and appear red in the nucleus. Type I and type II NBs are the biggest cells in the brain with a diameter of 10–15μm, whereas INPs and GMCs are much smaller and have a diameter of 4–7μm. In this and all the following figures all the images from the brain lobe are oriented in a way that the midline of the brain is to the right and anterior up. Please note that images are all from single focal slices and some type II NBs may not show in a single focal slice. The boundary between the central brain (CB) and the optic lobe (OL) are marked with dashed lines. Scale bars equal 50μm in (C, D, E-F) or 10μm in ($C_1$ and $D_1$). (A) A diagram shows the localization and cell type compositions of type II and type I NB lineages in *Drosophila* larval brains, with indicated molecular markers for type I and type II NBs, INPs, and GMCs. Type II NB lineages are localized in the central brain (CB) but not in the optic lobe (OL). (B) A schematic diagram depicts the progression of the development of a type II NB lineage. The expression patterns and their regulation of some key molecules involved in each step of type II NB lineage development are indicated (please see the text for the detailed description). (C-$C_1$") A wild type brain lobe has only eight type II NBs (arrows, only 7 are shown). ($C_1$-$C_1$") show an enlarged view of a single type II NB lineage highlighted with a dashed square in (C). Each type II NB lineages contains a single NB (arrows) and about 20–30 mINPs (arrowheads). (D-$D_1$") Knockdown of Six4 by *UAS-Sxi4 RNAi*$^{HM05254}$ driven by *pntP1-GAL4* leads to an increased number of type II NBs. ($D_1$-$D_1$") show an enlarged view of a single Six4 knockdown type II NB lineage that is highlighted with a dashed square in (D). Note this lineage contains three type II NBs that sit next to each other. Arrowheads in $D_1$-$D_1$" point to mINPs. (E-F) The supernumerary type II NB phenotype resulting from Six4 knockdown is rescued by expressing either *UAS-Six4ORF* (E) or *UAS-Six4*$^{HM05254Resis}$ (F). Note that mCD8-RFP driven by *pntP1-GAL4* sometimes has high expression in the glial sheath that wrap around both type I and type II NB lineages as shown in this and some other figures (e.g. open yellow arrow). (G) Quantifications of the number of type II NBs and mINPs with indicated genotypes. The number on top of each bar indicates the number of brains or lineages examined. ***, $p < 0.001$. NS, no significant.

and E(spl) proteins allows Pointed P1 (PntP1), which is required for type II NB specification and INP generation, to activate the expression of Earmuff (Erm) in imINPs [22–25]. Erm promotes INP maturation by inhibiting the activity and expression of PntP1 in imINPs as well as interacting with the epigenetic modifiers such as the SWI/SNF complex and HDAC3 [22,23,25,26]. Inhibition of PntP1's activity and expression in imINPs ensures imINPs become fate-committed mINPs instead of reverting to the type II NB identity. In addition to avoiding dedifferentiation, INPs also need to avoid premature differentiation and cell cycle exit in order to maintain their progenitor state. This requires not only the re-activation of self-renewing programs such as Dpn expression and activation of Notch signaling like in any NBs, but more importantly, the inhibition of the expression of homeodomain protein Prospero (Pros) in newly generated imINPs [17,27]. Nuclear Pros promotes the specification and cell cycle exit of GMCs in type I NB lineages [28]. In newly generated imINPs, the expression of Pros is normally suppressed by Buttonhead (Btd) and PntP1 [29–31]. The absence of Pros in imINPs ensures that INPs maintain the progenitor state and undergo self-renewing divisions instead of differentiation into GMCs and exiting cell cycle (Fig 1B) [27,29,30]. However, in spite of these important discoveries, the detailed molecular mechanisms underlying the regulation of dedifferentiation and differentiation of INPs by these molecules remain to be further elucidated. Furthermore, it is not clear whether there are any additional factors that could function together with these molecules to regulate the differentiation and/or dedifferentiation of INPs.

Six4 is a homeodomain transcription factor that belongs to the sine oculis homeobox (SIX) family, which is characterized by the presence of Six domains and homeodomains [32]. In both vertebrates and invertebrates, Six4 is required for patterning of the mesoderm and development of mesoderm-derived tissues, including muscles and gonads [33–36]. Six4 is also expressed in developing retina and neurogenic placodes where type II NBs originate [32,33,37], but its functions in these tissues remain to be deciphered. In this study, we identified Six4 as a novel protein regulating the homeostasis of INPs in type II NB lineages from an RNAi knockdown screen. We characterized the expression pattern of Six4 in the *Drosophila* larval CNS and examined its roles in regulating the development of INPs. We also investigated its functional and biochemical relationships with PntP1 and Erm in regulating INP development. Findings from this study suggest that Six4 is required for preventing both dedifferentiation and premature differentiation of INPs.

## Results

### Knockdown of Six4 results in supernumerary type II NBs

In order to identify genes that are involved in INP development, we have conducted an RNAi knockdown screen of transcription factors in type II NB lineages. In this screen, we labeled individual type II NB lineages with the expression of *UAS-mCD8-RFP* driven by *pntP1-GAL4* [24], which is also used to drive the expression of *UAS-RNAi* transgenes in type II NB lineages. Meanwhile, all the NBs and mature INPs were labeled by *E(spl)mγ-GFP* [16]. With these markers, we could easily identify the type II NB (the biggest cell expressing *E(spl)mγ-GFP*), imINPs (next to the type II NBs without the expression of *E(spl)mγ-GFP*), and mINPs (expressing *E(spl)mγ-GFP* but smaller than the NB) in individual type II NB lineages and detect changes in the number of these cell types in live larval brains (Fig 1A). In wild type larvae, each brain lobe contains 8 type II NBs and each type II NB lineage contains a single NB, 4–5 imINPs and 20–30 mINPs [7–9,24] (Fig 1A, 1C–1C₁" and 1G). From this screen, we found that, the number of type II NBs was increased to about 16/brain lobe when the *UAS-Six4 RNAi*$^{HM05254}$ transgene was expressed in type II NB lineages (Fig 1D–1D₁" and 1G). Similar supernumerary type II NB phenotypes were observed when three additional independent *UAS-Six4 RNAi* lines were used to knock down Six4 (S1B–S1D₁ and S1H Fig). Most of these supernumerary type II NBs resulting from Six4 knockdown established their own independent lineages. However, about 4% of single isolated lineages had two or more type II NBs at the 3$^{rd}$ instar larval stages (Figs 1D₁–1D₁" and S1B₁ and S1D₁). Co-existence of multiple type II NBs in single lineages indicates that these extra type II NBs could be just generated during the lineage development at late larval stages and have not established their own independent lineages yet. However, we did not observe dramatic changes in the average number of mINPs in individual lineages (Figs 1G and S1H). To confirm that the supernumerary type II NB phenotypes were caused by Six4 knockdown, we performed rescue experiments. Indeed, the supernumerary type II NB phenotype resulting from the expression of *UAS-Six4 RNAi*$^{HM05254}$ was fully rescued by expressing Six4 using either its wild type sequence (*UAS-Six4ORF-HA*, simplified as *UAS-Six4ORF* thereafter) [38] or an RNAi-resistant sequence (*UAS-Six4*$^{HM05254Resis}$) (Fig 1E–1G). The rescue of the knockdown phenotypes is unlikely due to dilution of GAL4 activity and reduced efficiency of RNAi knockdown because including an additional copy of the *UAS-mCD8-RFP* transgene did not affect the supernumerary type II NB phenotypes resulting from Six4 knockdown (S1A and S1H Fig). These results indicate that the supernumerary type II NB phenotype is indeed caused by knockdown of Six4. Therefore, Six4 is required to prevent the generation of supernumerary type II NBs. Since knockdown of Six4 with the *UAS-Six4 RNAi*$^{HM05254}$ (simplified as *UAS-Six4 RNAi* thereafter) transgene gave the strongest phenotype, we used this line for the rest of study.

Since most supernumerary type II NBs established independent lineages when Six4 was knockdown, this raised a question of whether most supernumerary type II NBs were generated early during development. To address this question, we then tried to express *UAS-Six4 RNAi* only after larval hatching by combining the *pntP1-GAL4* driver with the temperature sensitive GAL80 under the control of tubulin promoter (*tub-GAL80*$^{ts}$). We found that knocking down Six4 after larval hatching led to much weak supernumerary type II NB phenotypes (S1E–S1G and S1I Fig). With two copies of *UAS-Six4 RNAi*, we observed one to three extra type II NBs in 88% of brain lobes with an average of 9 type II NB per brain lobe (S1G and S1I Fig). With one copy of *UAS-Six4 RNAi*, the phenotype became even weaker. Only 35% of brain lobes contain one or two extra type II NBs with an average of 8.4 type II NBs per brain lobe (S1E–S1F₂ and S1I Fig). Although the weak phenotype could be potentially in part due to reduced efficiency of Six4 knockdown because of the inclusion of *tub-GAL80*$^{ts}$ with the driver and a shorter

duration of the expression of the RNAi transgene, these results indicate that the majority of the supernumerary type II NBs resulting from Six4 knockdown were probably generated at embryonic stages and possibly early larval stages as well.

## Expression of Six4 in *Drosophila* larval type II neuroblast lineages

As a first step to investigate why knockdown of Six4 led to supernumerary type II NBs, we examined if Six4 is specifically expressed in type II NB lineages and which cell types express Six4. Unfortunately, immunostaining using the existing Six4 antibody [37] did not give any detectable signals in larval brains. Therefore, we used a GFP reporter line of six4 (*six4-GFP*) from the Model Organism Encyclopedia of Regulatory Networks (modERN) Project to examine the expression of Six4. The *Six-GFP* line was generated by inserting the superfolder GFP (sfGFP) sequence at the 3'-end of the *Six4* coding sequence in the P[acman]CH322-192H07 BAC clone that contains a ~9.4kb fragment upstream of the transcription start site of s*ix4* (Fig 2A) [39]. Therefore, the expression of Six4-GFP fusion protein should reflect the expression patterns of endogenous Six4. We examined Six4-GFP expression from 1st through late 3rd instar larval stages by immunostaining. We found that the Six4-GFP is consistently expressed in all type II NB lineages throughout larval development (Figs 2B–2B$_1$" and S2C–S2C', only images from third instar larval brains are shown). In each type II NB lineage, Six4-GFP is expressed in the type II NB, imINPs, young mINPs that are close to imINPs, and a few GMCs next to the young mINPs. The expression of Six4-GFP is the strongest in the NB but becomes progressively weaker in the daughter cells with the increase of their distance from the NB (Fig 2B–2B$_1$"). The expression of Six4-GFP in type II NB lineages was abolished when *UAS-Six4 RNAi*$^{HM05254}$ was expressed in type II NB lineages (Fig 2C–2C"), providing additional evidence to support that the supernumerary type II NB phenotype resulting from the expression of *UAS-Six4 RNAi* transgenes was caused by the loss of Six4. In addition to type II NB lineages, Six4-GFP is also consistently expressed in 5–6 type I NB lineages in the ventral nerve cord (VNC) and 10–12 type I NBs in each larval brain lobe (S2A–S2C' Fig). The expression pattern of Six4-GFP is consistent with recent RNAseq data showing that Six4 is enriched in type II NBs [40]. The expression of Six4-GFP supports the role of Six4 in regulating type II NB lineage development as demonstrated by the RNAi knockdown phenotypes.

## Six4 likely acts in imINPs to prevent dedifferentiation of imINPs

The generation of supernumerary type II NBs could potentially result from dedifferentiation of imINPs [9,29,41,42], defects in asymmetric cell division of type II NBs [43,44], or aberrant specification of NBs from the neuroectoderm during embryonic stages [37]. Since we drove the expression of *UAS-Six4 RNAi* transgenes with *pntP1-GAL4*, which like endogenous PntP1 proteins is only expressed in type II NBs at embryonic stages but not in type I NBs or the neuroectoderm before type II NBs are specified [45,46], it is unlikely that Six4 knockdown would affect the specification of type II NBs from the neuroectoderm during the embryonic stages. Furthermore, the coexistence of multiple type II NBs in single Six4 knockdown lineages, although at a low rate, also suggests that the supernumerary type II NBs could be generated during the lineage development after they are specified. Therefore, the supernumerary type II NBs resulting from Six4 knockdown are probably generated from either dedifferentiation of imINPs or defects in asymmetric cell division of type II NBs.

To determine whether Six4 knockdown affected the asymmetric division of type II NBs, we then examined the distribution of the apical protein atypical protein kinase C (aPKC) and the basal protein Miranda (Mira) during the division of type II NBs. We stained for Phospho-histone H3 (PH3), which is coupled with chromosome segregation and condensation [47], to

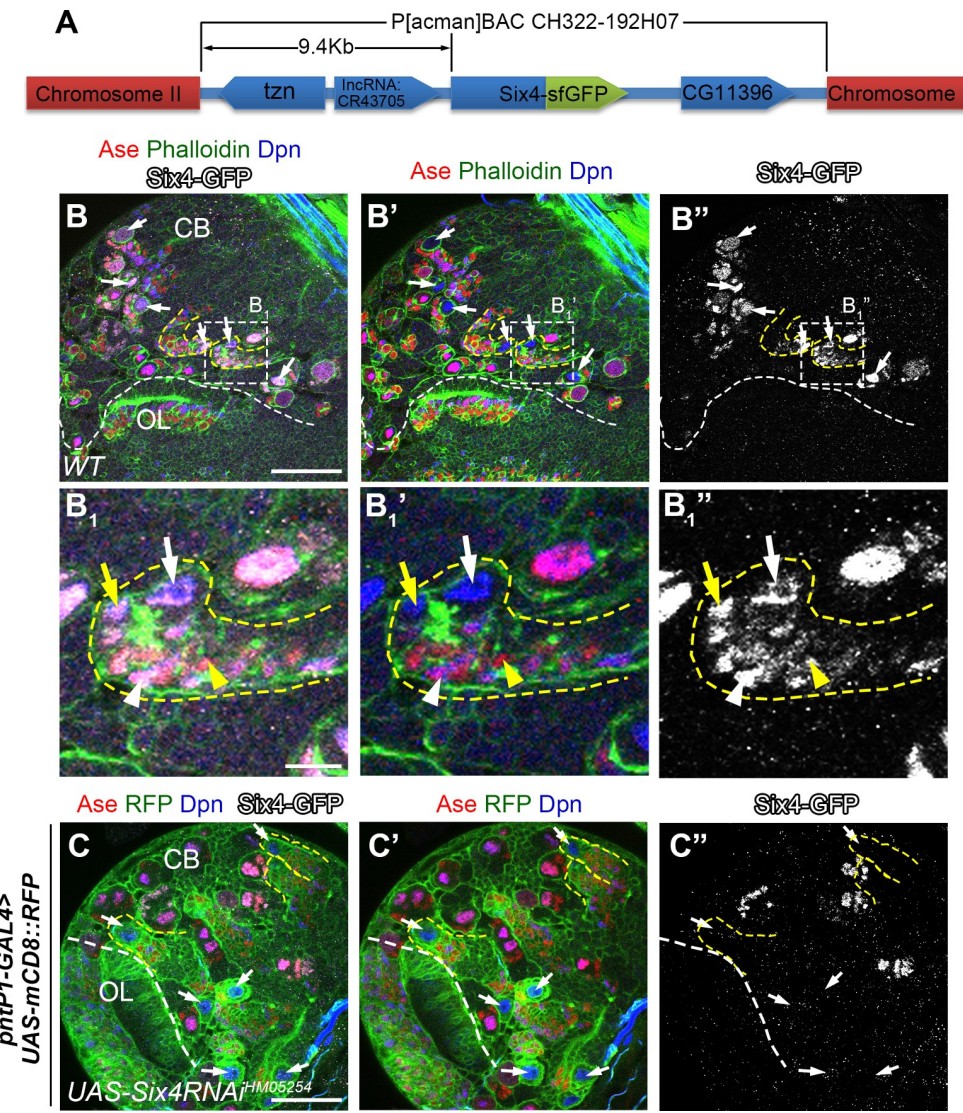

**Fig 2. Six4 is expressed in type II NB lineages.** (A) A schematic diagram for *Six4-GFP* line shows in-frame fusion of GFP to the C-terminus of the Six4 gene, which is under the control of a 9.4kb native promoter/enhancer fragment in BAC clone P[acman]CH322-192H07. (B-B₁") Six4-GFP is expressed in type II NB lineages (e.g. outlined by dashed lines) in a 3rd instar larval brain lobe. The brain was counterstained with Phalloidin that labels both type I and type II NB lineages and antibodies against Dpn, Ase, and GFP. (B₁-B₁") An enlarged view of Six4 expression in a single type II NB lineage from the area highlighted with a dashed square in (B-B"). White arrows: NBs; yellow arrows: imINPs; white arrowheads: young mINPs; yellow arrowheads: GMCs. (C-C") Expression of *UAS-Sxi4 RNAi^HM05254* driven by *pntP1-GAL4* abolishes Six4-GFP expression in all type II NB lineages, which are labeled by mCD8-RFP. Some type II NB lineages are outlined by dashed lines as an example. Scale bars equal 50μm in (B, C) or 10μm in B₁.

determine the phases of the cell cycle. Our results showed both Mira and aPKC in Six4 knock-down NBs exhibited the same distribution patterns as in wild type NBs during the division. At metaphase, Mira was appropriately segregated to basal cortex, whereas aPKC was segregated to the apical cortex (S3A1–S3B1' Fig). At anaphase and telophase, Mira was segregated into the future imINPs on the cell cortex but not in NBs, whereas aPKC persisted weakly only on the apical cortex of NBs. Furthermore, two future daughter cells displayed unequal sizes (S3A2–S3B2' Fig). These results demonstrate that the supernumerary type II NB phenotype

resulting from Six4 knockdown is not caused by asymmetric cell division defects of type II NBs.

To test if the supernumerary type II NBs were generated by dedifferentiation of imINPs, next we used *erm-GAL4(II)* and/or *erm-GAL4 (III)* to knock down Six4 in imINPs specifically. *erm-GAL4(II)* is expressed in both Ase⁻ and Ase⁺ imINPs except the newly born imINPs, whereas *erm-GAL4(III)* is expressed slightly later, mainly in Ase⁺ imINPs [48,49]. We found that although knockdown of Six4 in imINPs with one copy of *UAS-Six4RNAi* driven by *erm-GAL4(II)* alone or together with *erm-GAL4(III)* did not yield any phenotypes (S4A–S4C Fig), expression of two copies of *UAS-Six4 RNAi* driven by two copies of *erm-GAL4(II)* consistently led to generation of 1–2 extra type II NBs per brain lobe with one lineage containing more than one type II NBs per brain lobe on average, which never occurred when two copies of *UAS-mCherry RNAi* was expressed (Fig 3A–3C" and 3F). Although the phenotype is weak, the consistent generation of extra type II NBs and the existence of multiple type II NBs in single lineages indicate that imINPs could dedifferentiate to adopt the type II NB fate when Six4 was knocked down in imINPs. The weak phenotype could be due to insufficient knockdown of Six4 in imINPs as indicated by immunostaining (S4D–S4E" Fig). Nevertheless, we were able to observe much stronger supernumerary type II NB phenotypes when Six4 was knocked down in imINPs in *erm²*/+ heterozygous mutant background (also see below, Fig 3D–3E" and 3G). Accordingly, the average number of mINPs per NB was reduced by 32% (Fig 3H). These data further support that knockdown of Six4 in imINPs could lead to dedifferentiation of imINPs (Fig 3I and 3J).

## Six4 prevents dedifferentiation of imINPs likely by inhibiting the expression and activity of PntP1

Our previous studies as well as others have shown that maturation of imINPs requires suppression of PntP1's activity and expression in imINPs by Erm (Fig 1B) [22,23,25]. Thus, to investigate why knockdown of Six4 led to dedifferentiation of imINPs, we examined if Six4 regulates the expression of PntP1 and/or Erm in imINPs. We first examined the expression of PntP1 in Six4 knockdown type II NB lineages. In wild type type II NB lineages, PntP1 was strongly expressed in type II NBs, Dpn⁻ E(spl)mγ⁻ imINPs, but absent in Dpn⁺ E(spl)mγ⁺ mINPs (Fig 4A–4A" and 4I) [24]. In Six4 knockdown type II NB lineages, the staining intensities of PntP1 in the NB and imINPs were comparable to those in the wild type lineages. However, PntP1 remained expressed in newly generated E(spl)mγ⁺ mINPs right next to imINPs in the Six4 knockdown type II NB lineages although the expression level was lower than that in imINPs (Fig 4B–4B" and 4I), indicating that the suppression of PntP1 expression in mINPs was delayed in the absence of Six4. Since Erm represses the expression of PntP1 [23,25], we then examined if the persistence of PntP1 in the newly generated mINPs was caused by decrease in Erm expression. However, we did not observe obvious changes in Erm expression in Six4 knockdown imINPs compared with that in the wild type (Fig 4C–4D' and 4J). These results suggest that, like Erm, Six4 is also required to suppress PntP1 expression in INPs and the dedifferentiation of imINPs in Six4 knockdown type II NB lineages is likely due to delayed suppression of PntP1 expression in INPs but not changes in Erm expression.

To further prove that Six4 inhibits the expression of PntP1, we examined if Six4 overexpression would suppress PntP1 expression in type II NB lineages and subsequently lead to transformation of type II NB lineages into type I-like NB lineages and/or loss of INPs. Indeed, overexpression of Six4 driven by *pntP1-GAL4* resulted in transformation of 25% of type II NB lineages into type I-like lineages as indicated by the ectopic expression of Ase in the NB and the presence of 4–6 GMC-like cells but no mINPs beside the NBs. In addition, about 12.5% of

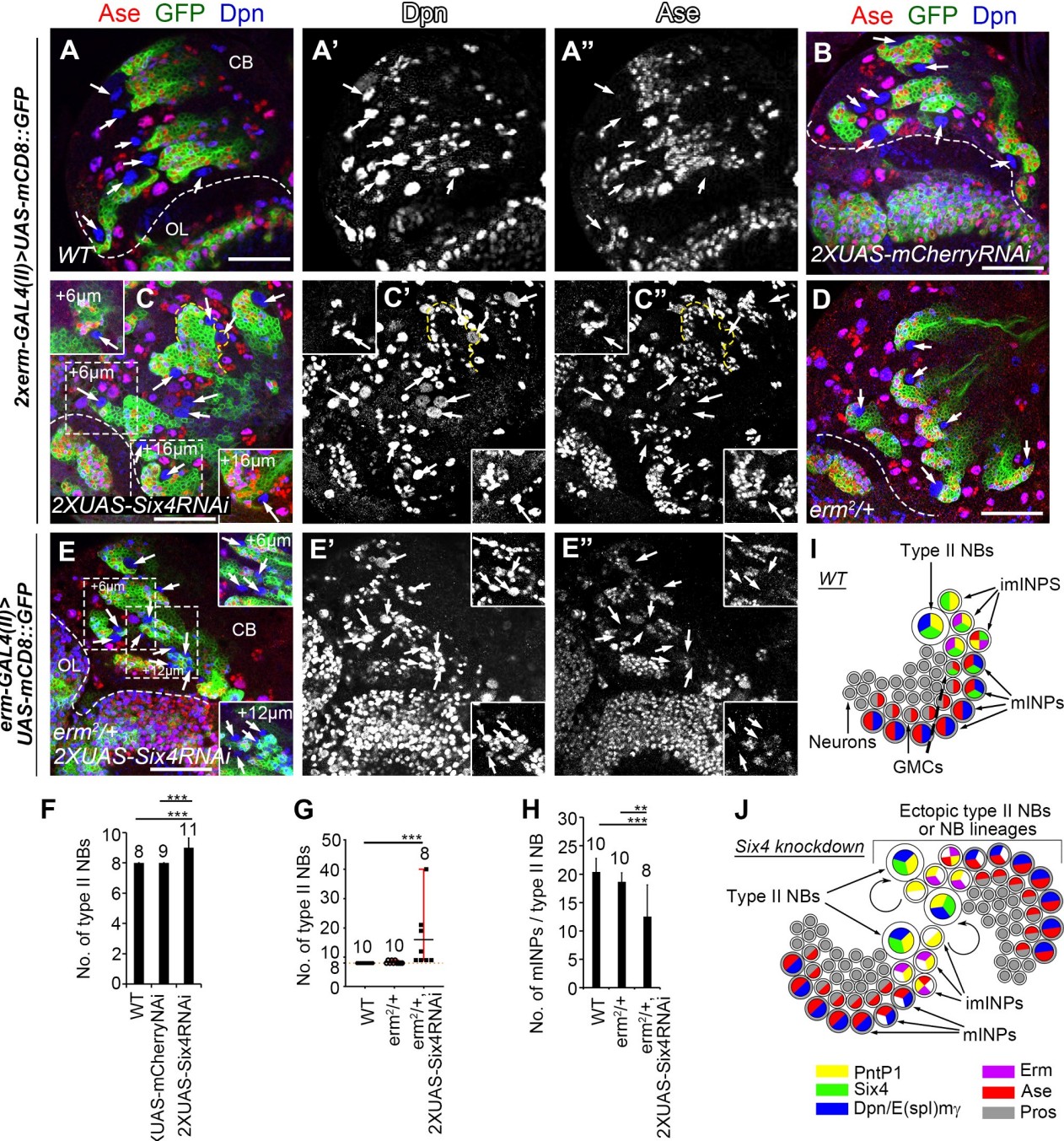

**Fig 3. Knockdown of Six4 in imINPs leads to an increased number of type II NBs.** In all images, type II NB lineages were labeled with mCD8-GFP driven by *erm-GAL4 (II)* and counterstained with anti-Dpn and anti-Ase antibodies. Arrows point to type II NBs that express Dpn but not Ase. Note that type II NBs are not labeled by mCD8-GFP driven by *erm-GAL4* (II). Dashed lines demarcate the boundary between the central brain (CB) and the optic lobe (OL). Scale bars equal 50μm. (A-A") A wild type brain lobe contains eight type II NBs. (B) Expression of two copies of *UAS-mCherry RNAi* driven by two copies of *erm-GAL4(II)* does not generate any ectopic type II NBs. (C-C") Expressing two copies of *UAS-Six4 RNAi* driven by two copies of *erm-GAL4(II)* results in extra type II NBs. Insets show two additional type II NBs at different focal planes from the areas highlighted with dashed squares. A single lineage with two type II NBs is outlined by a yellow dashed line. (D) An *erm²/+* heterozygous mutant brain lobe has eight type II NBs. (E-E") Knockdown of Six4 with two copies of *UAS-Six4 RNAi* in *erm²/+* heterozygous mutant background results in generation of more extra type II NBs. Insets show additional type II NBs at different focal planes from the areas highlighted with dashed squares. (F-G) Quantifications of the number of type II NBs in the brains with indicated genotypes. The number on top of each bar represents the number of brains examined. ***, $p < 0.001$. (H) Quantifications of the average number of mINPs per NB in the brains with indicated genotypes. The number on top of each bar represents the number of brains examined. **, $p < 0.01$. ***, $p < 0.001$. (I) A schematic diagram shows the expression pattern of Six4 (green) in a type

II NB lineage. (J) A schematic diagram shows that knockdown of Six4 leads to dedifferentiation of imINPs into type II NBs that establishes independent lineages.

type II NBs failed to produce any mINPs even though their identity was maintained correctly as indicated by the absence of Ase. The average number of INPs per type II NB lineage was reduced to about 10 when Six4 was overexpressed, whereas a wild type type II NB lineage contains an average of 23 INPs (Figs 4E–4F$_1$" and 4K–4M and S5L). Accordingly, we found that PntP1 expression in the Ase$^+$ type II NBs was drastically reduced by 75%. Even in the Ase$^-$ type II NBs, the PntP1 expression was also reduced by 30% (Fig 4G–4H" and 4N). Similar suppression of PntP1 was also observed in Six4 overexpressing imINPs (Figs 4G–4H" and 4O and S5L).

Our previous studies have shown that loss of INPs resulting from Buttonhead (Btd) knockdown could lead to loss of PntP1 in type II NBs and rescue of INPs by reducing Pros expression could restore the expression of PntP1 in Btd knockdown type II NBs, suggesting that maintaining PntP1 expression in type II NBs may require a feedback signal from INPs and the loss/reduction of PntP1 expression could be a secondary effect due to loss of INPs [30]. To investigate whether the loss/reduction of PntP1 in Six4 overexpression type II NBs is a secondary effect of the loss of INPs, we then examined if the loss/reduction of PntP1 expression could be restored by rescuing the loss of INPs. To this end, we first wanted to determine if ectopic Pros expression in imINPs was similarly responsible for the loss of mINPs resulting from Six4 overexpression. As observed in PntP1 knockdown type II NB lineages [29], we found that Pros was ectopically expressed in imINPs even in the lineages without ectopic Ase expression in the NBs when Six4 was overexpressed (S5A–S5B" Fig). Furthermore, the loss of mINPs resulting from Six4 overexpression was fully rescued in *pros$^{17}$*/+ heterozygous mutant larvae (S5C–S5D' and S5G and S5H Fig), demonstrating that the loss of mINPs was also due to ectopic Pros expression in imINPs and subsequent premature differentiation of INPs into GMCs. Then we examined if PntP1 expression in the NB was still suppressed by Six4 overexpression when mINPs were rescued in *pros$^{17}$*/+ heterozygous mutants. Our immunostaining showed that when Six4 was overexpressed in *pros$^{17}$*/+ type II NB lineages, PntP1 expression was still reduced by 30% in Ase$^-$ type II NBs even though mINPs were rescued. Furthermore, 25% of type II NBs still ectopically expressed Ase and the expression of PntP1 in these Ase$^+$ type II NBs was still reduced by 42% (S5E–S5F", S5I and S5J Fig). Similar suppression of PntP1 was also observed in Six4 overexpressing imINPs (S5E–S5F" and S5K Fig). Therefore, unlike in Btd knockdown type II NBs, rescuing mINPs did not fully restore the expression of PntP1 in Six4 overexpression type II NBs, suggesting that although the loss of INPs does contribute to the reduction of PntP1 expression to a certain extent, it is not the major contributing factor to the loss of PntP1 expression in Six4 overexpression type II NBs. Rather, our results demonstrate that Six4 overexpression indeed suppresses PntP1 expression in type II NBs either directly or indirectly. Furthermore, the ectopic expression of Ase in type II NBs with only 42% of reduction in PntP1 expression suggests that PntP1 activity might be inhibited as well by Six4 overexpression.

In order to more directly assess if Six4 antagonizes the function of PntP1, we next examined 1) if transformation of type I NB lineages into type II-like NB lineages induced by PntP1 misexpression could be inhibited by concomitant expression of Six4; and 2) if Erm expression in imINPs would still be abolished by Six4 overexpression if PntP1 expression was maintained in type II NB lineages. We used the pan-NB driver *insc-GAL4* to drive the expression of Six4 and/or PntP1 in both type I and type II NBs. For examining the transformation of type I NB lineages into type II-like NB lineages, we focused our phenotypic analyses in the VNC, which

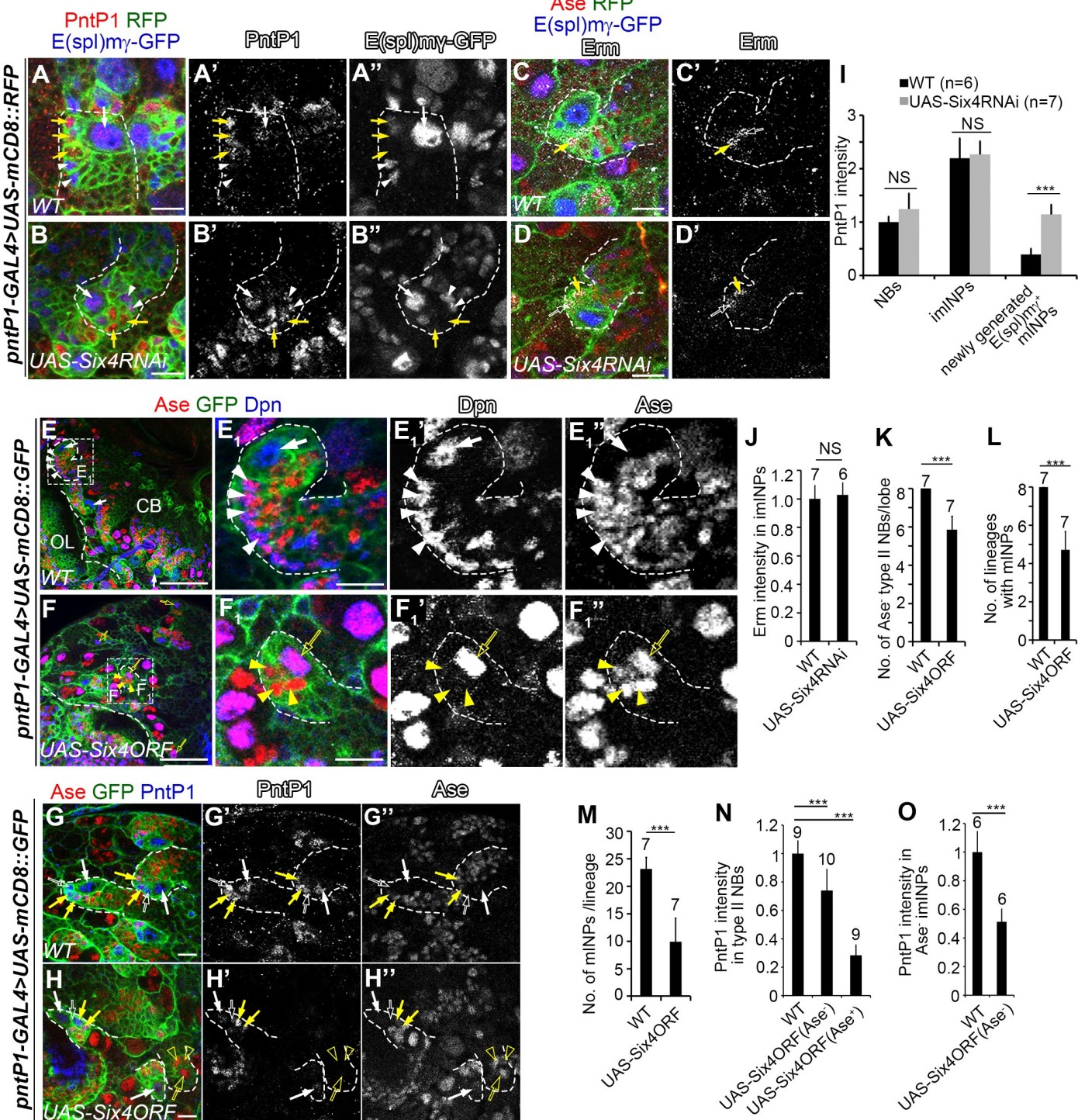

**Fig 4. Six4 inhibits PntP1 expression in type II NB lineages.** In all images, type II NB lineages were labeled with mCD8-RFP or mCD8-GFP driven by *pntP1-GAL4* and counterstained with a combination of various antibodies as indicated. Some type II NB lineages are highlighted with dashed lines as an example. Scale bars equal 10μm in (A-D', E1, F1, G-H") or 50μm in (E, F). (A-A") In wild type type II NB lineages, PntP1 is expressed in type II NBs (white arrows) and imINPs (E(spl)mγ⁻, yellow arrows), but is absent in newly generated E(spl)mγ⁺ mINPs (white arrowheads). (B-B") PntP1 is expressed in newly generated E(spl)mγ⁺ mINPs (white arrowheads) in addition to type II NBs (white arrows) and imINPs (yellow arrows) in Six4 knockdown type II NB lineages. (C-C') Erm is expressed in Ase⁻ (open white arrows) and Ase⁺ (yellow arrows) imINPs in wild type type II NB lineages (dashed lines). (D-D') Erm remains expressed in Ase⁻ (open white arrows) and Ase⁺ (yellow arrows) imINPs in Six4 knockdown type II NB lineages. (E-E1") Ase is absent in wild type type II NBs (arrows) and there are multiple mINPs (arrowheads) in individual lineages. (E1-E1") show an enlarged view of the lineage highlighted with a dashed square in (E). (F-F1") Six4 overexpression leads

to ectopic Ase expression in the NBs (open yellow arrows) and loss of mINPs in a subset of type II NB lineages. Instead, only GMCs (yellow arrowheads) are generated in these lineages as shown in the enlarged view ($F_1$-$F_1$”) of the lineage highlighted with a dashed square in (F). (G-H”) Six4 overexpression (H-H”) leads to a significant reduction in PntP1 expression in Ase$^-$ type II NBs (white arrows), Ase$^-$ (open white arrows) and Ase$^+$ (yellow arrows) imINPs compared to the wild type control (G-G’). PntP1 is almost abolished in Ase$^+$ type II NBs (open yellow arrows) and their associated Ase$^+$ progeny (open yellow arrowheads) in the Six4 overexpressing brain (H-H”). (I-M) Quantifications of PntP1 (I) or Erm (J) staining intensity, the number of Ase$^-$ type II NBs (K), the number of type II NB lineages with mINPs (L), and the number of mINPs/lineage (M) in wild type, Six4 knockdown or Six4 overexpressing brains. The number on top of each bar indicates the number of cells or lineages examined. $^{***}P<0.001$. NS, no significant. (N-O) Quantifications of PntP1 intensity in type II NBs (N) and imINPs (O) in wild type and Six4 overexpressing type II NB lineages with or without ectopic Ase expression in the NBs. The number on top of each bar indicates the number of cells or lineages examined. $^{***}$, $p < 0.001$.

contains only type I NB lineages. PntP1 misexpression resulted in suppression of Ase expression in 90% type I NBs and generation of Ase$^+$ Dpn$^+$ mINP-like cells in about 15% of type I NB lineages in the VNC (Fig 5A–5B’, 5I and 5J), which is consistent with previous studies showing that PntP1 misexpression transforms type I NB lineages into type II-like NB lineages [24,30]. Type I NB lineages developed normally when Six4 was misexpressed (Fig 5C–5C’, 5I and 5J). However, when Six4 and PntP1 were misexpressed simultaneously in type I NBs, the suppression of Ase expression was observed in only 15% of type I NBs and no INP-like cells were generated (Fig 5D–5D’, 5I and 5J), suggesting that PntP1 could no longer function effectively to suppress Ase and induce the generation of INP-like cells when Six4 was simultaneously expressed. Consistently, we found that Erm expression in imINPs was abolished when Six4 was overexpressed either alone or together with PntP1 in type II NBs (Fig 5E–5H’ and 5K), indicating that even if PntP1 expression was maintained in type II NBs, it could no longer activate Erm expression when Six4 was overexpressed. The inhibition of PntP1-induced transformation of type I NB lineages into type II-like NB lineages and the suppression of PntP1-activated Erm expression in imINPs suggest that Six4 antagonizes the activity of PntP1.

Taken together, the above results demonstrate that like Erm, Six4 can not only suppress PntP1 expression but also antagonize PntP1's activity. The inhibition of PntP1's expression and activity in imINPs by Six4 ensures imINPs differentiate properly to become fate committed mINPs.

### Six4 and Erm genetically interact in preventing dedifferentiation of imINPs

Similar functions of Six4 and Erm in preventing dedifferentiation of imINPs by inhibiting the expression and activity of PntP1 led us to ask what is the functional relationship between Six4 and Erm. Do they function in the same pathway or in two parallel independent pathways? Our results that Six4 knockdown did not affect the expression of Erm in imINPs (Fig 4C–4D’ and 4J) and Six4 overexpression abolished Erm expression (Fig 5G–5H’ and 5K) indicate that at least Six4 does not function upstream to activate Erm expression. To test if Six4 could function downstream of Erm, we examined Six4-GFP expression while knocking down Erm or its activator PntP1. However, we did not observe obvious changes in Six4-GFP expression in individual type II NB lineages when Erm or PntP1 was knocked down (S6 Fig). Therefore, it seems that there is no cross-regulatory relationship between Six4 and Erm/PntP1.

To further determine the functional relationship between Six4 and Erm, then we performed the following genetic interaction tests. First, we examined if removing one wild type copy of *erm* would enhance the Six4 knockdown phenotypes. To this end, we knocked down Six4 in the type II NB lineage with *pntP1-GAL4* or specifically in imINPs with *erm-GAL4 (II)* in *erm$^2$/* + heterozygous mutant background. Type II NB lineages developed normally in *erm$^2$/+* heterozygotes except that there was one extra type II NB in about 30% of brain lobes when *erm-GAL4 (II)* was used to drive the expression of mCD8-GFP in type II NB lineages (Fig 3D and

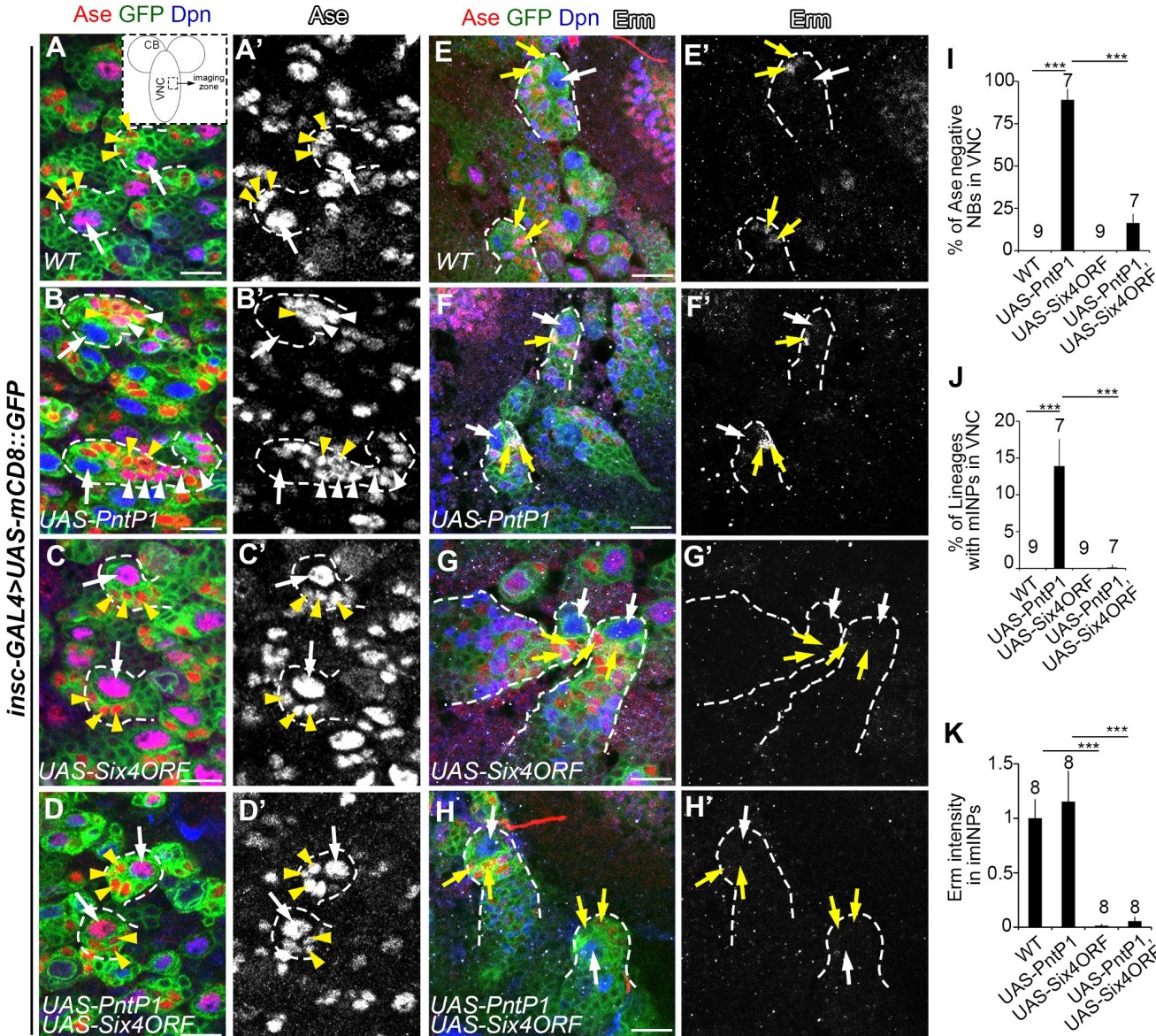

**Fig 5. Six4 antagonizes PntP1's activity.** Individual type I NB lineages (A-D) or type II NB lineages (E-H) were labeled with mCD8-GFP driven by *insc-GAL4* and counterstained with antibodies against Dpn, Ase, and/or, Erm. Scale bars equal 10μm. (A-A') All wild type type I NBs (white arrows) in VNC are Ase$^+$ Dpn$^+$ and produce Ase$^+$ GMCs (yellow arrowheads). The diagram in the inset indicates the area imaged in the VNC. (B-B') Misexpression of PntP1 driven by *insc-GAL4* in type I NBs suppresses Ase expression in type I NBs (white arrows) and results in generation of INP-like cells (white arrowheads), which in turn produce GMCs (yellow arrowheads). (C-C') Six4 misexpression in type I NBs does not affect Ase$^+$ expression (e.g. arrows) or GMC generation (yellow arrowheads). (D-D') Simultaneous misexpression of Six4 and PntP1 does not suppress Ase expression in type I NBs (arrows) or induce INP-like cells. Instead, only GMCs (yellow arrowheads) are generated. (E-F') Erm is expressed in imINPs (yellow arrows) of wild type (E and E') or PntP1 overexpressing (F and F') type II NB lineages. White arrows: type II NBs. (G-H') Erm expression is not detected in imINPs (yellow arrows) when Six4 is overexpressed alone (G and G') or together with PntP1 (H and H'). White arrows: type II NBs. (I-K) Quantifications of the percentage of Ase⁻ NBs in the VNC (I), the percentage of NB lineages with ectopic INP-like cells in the VNC (J), and the staining intensity of Erm in imINPs (K) in the wild type or PntP1 or/and Six4 mis-/over-expressing larvae. The numbers on top of each bar represent the numbers of VNCs (I-J) or the number of imINPs (K) examined. ***, $p < 0.001$.

3G). Significantly, the average number of type II NBs was increased to 20/brain lobe when *pntP1-GAL4* was used to knock down Six4 in *erm²/+* heterozygotes, which was over 30% of increase compared to the knockdown of Six4 in the wild type background (Fig 6A–6C' and

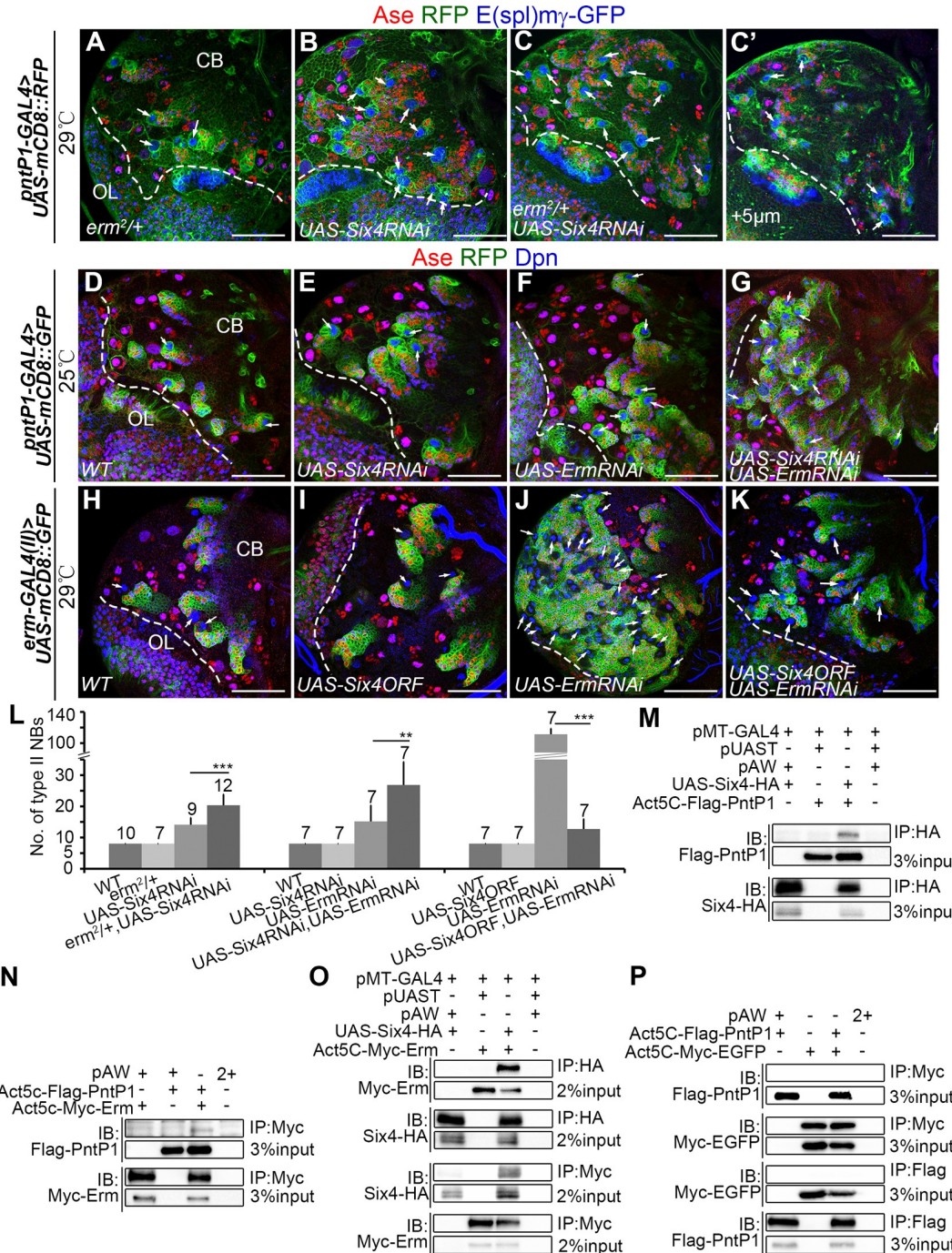

**Fig 6. Six4 functions together with Erm to prevent dedifferentiation of imINPs by forming a complex with PntP1.** Type II NB lineages are labeled by mCD8-RFP (A-C') or mCD8-GFP (D-G) driven by *pntP1-GAL4* or mCD8-GFP driven by *erm-GAL4 (II)* (H-K). The brains are counterstained with antibodies against Ase, Dpn, RFP or GFP. Arrows point to type II NBs. Scale bars equal 50μm. (A-C') There are only 8 type II NBs/brain lobe in the *erm²/+* heterozygous mutant (A) animals. Knockdown of Six4 with *pntP1-GAL4* results in an increased number of type II NBs (B), which was further increased when Six4 is knocked down in the *erm²/+* heterozygous mutant background (C-C'). (C-C') are two different focal slices from the same brain. (D-G) A larval brain expressing *UAS-Six4 RNAi* driven by *pntP1-GAL4* at 25˚C (E) has only 8 type II NBs as in the wild type brain (D). Knockdown of Erm at 25˚C results in a slight increase in the number of type II NBs (F). However, simultaneous knockdown of Six4 and Erm at 25˚C leads to a synergistic increase in the number of type II NBs (G). (H-K) The number of type II NBs remains the same as in the wild type (H) when Six4 is overexpressed alone in imINPs (I). Knockdown of Erm in imINPs at 29˚C results in supernumerary type II NBs (J), which is largely suppressed by simultaneous

overexpression of Six4 in imINPs (K). (L) Quantifications of the number of type II NBs in the brains with indicated genotypes. The number on top of each bar represents the number of brain lobes examined. ***, $p < 0.001$; **, $p < 0.01$. (M-P) Biochemical interactions between Flag-PntP1and Six4-HA (M), between Flag-PntP1 and Myc-Erm (N), between Six4-HA and Myc-Erm (O), are detected by transfecting S2 cells with their corresponding expression constructions followed by co-IP and Western blotting analyses, but not between Flag-PntP1 and Myc-EGFP under the same condition (P).

6L). Similarly, the supernumerary type II NB phenotypes resulting from Six4 knockdown in imINPs was significantly enhanced. As shown above, knockdown of Six4 in imINPs with two copies of *UAS-Six4 RNAi* in the wild type background only led to one extra type II NB per brain lobe on average. However, when similar knockdown was performed in *erm²/+* heterozygotes, there were 16 type II NBs/brain lobe on average, ranging from 9 to 40 per brain lobe (Fig 3D–3E" and 3G). Second, we tested if knocking down Six4 and Erm simultaneously would have synergistic effects on the generation of supernumerary type II NBs. Since Erm knockdown alone with *pntP1-GAL4* at 29˚C generated strong supernumerary type II NB phenotypes, making it difficult to assess the potential synergistic effect of double knockdown of Six4 and Erm, we performed RNAi knockdown at 25˚C to reduce the RNAi knockdown efficiency. At 25˚C, knockdown of Erm alone led to an average of 15 type II NBs per brain lobe, whereas knockdown of Six4 alone did not generate any extra type II NBs. Remarkably, double knockdown of Six4 and Erm resulted in 26 type II NBs per brain lobe on average, which is over 70% of increase compared to the Erm knockdown alone (Fig 6D–6G and 6L), suggesting that simultaneous knockdown of Six4 and Erm synergistically but not just additively enhances the supernumerary type II NB phenotype. These results suggest that although they do not regulate each other's expression, Six4 and Erm genetically interact and likely function in the same pathway to prevent dedifferentiation of imINPs. The synergistic enhancement of the supernumerary type II NB phenotype in Six4 and Erm double knockdown brains also indicates that Six4 and Erm may have partial functional redundancy.

In order to further confirm if Six4 and Erm function partially redundantly in imINPs, we examined if overexpressing Six4 could rescue the supernumerary type II NB phenotype resulting from Erm knockdown in imINPs. Knocking down Erm alone in imINPs with *erm-GAL4 (II)* at 29˚C led to an average of 111 type II NBs/brain lobe with 100% of penetrance of the phenotype, whereas expression of *UAS-Six4ORF* driven by *erm-GAL4(II)* in imINPs did not affect the number of type II NBs or the generation of INPs (Fig 6H–6J and 6L). Interestingly, the supernumerary type II NB phenotype resulting from Erm knockdown was largely rescued by overexpressing Six4 in imINPs. There were only 12 type II NBs/brain lobe on average and about 10% of brain lobes had no extra type II NBs when Six4 was overexpressed in Erm knockdown imINPs (Fig 6K and 6L). These results demonstrate that Six4 can functionally substitute Erm to prevent dedifferentiation of imINPs if its expression level is elevated.

## Six4, Erm, and PntP1 likely form a trimeric complex to inhibit PntP1 activity

Our results showed Six4 and Erm both antagonize the function of PntP1 that likely acts as a transcriptional activator [24,50]. There are two potential mechanisms underlying the inhibition of PntP1's function by Six4 or Erm. One is that Six4 or/and Erm binds to PntP1, which could prevent PntP1 from binding to its target genes or inhibit its transcriptional activity. Alternatively, Six4 and Erm function as transcriptional repressors to suppress the expression of PntP1 target genes in imINPs by binding to the same target genes without physically interacting with PntP1. To distinguish between these two possibilities, we performed biochemical interaction tests. We co-expressed HA-tagged Six4 or Myc-tagged Erm together with Flag-

tagged PntP1 in *Drosophila* S2 cells and examined their potential interactions by co-immuno-precipitation (co-IP). Our co-IP results showed that both Six4 and Erm could bind to PntP1 when they were co-expressed with PntP1 (Fig 6M and 6N). We excluded the possibility that these interactions in S2 cells were artificial due to high expression levels of these proteins because no interactions were detected when Myc-tagged EGFP was similarly co-expressed with Flag-PntP1 in S2 cells (Fig 6P). Since Six4 and Erm likely function in the same pathway based on our genetic interaction results but they do not regulate each other's expression, one possibility could be that Six4 and Erm function in the same complex together with PntP1. The presence of all these three proteins, Six4, Erm, and PntP1, may enhance their interactions and increase the stability of the complex, which could lead to stronger inhibition of PntP1's activity. To test this possibility, we first tested if Six4 and Erm could also bind to each other and co-exist with PntP1 in the same complex. Indeed, our co-IP results showed that Six4-HA and Myc-Erm could be co-IPed when they were co-expressed in S2 cells (Fig 6O). When Six4-HA, Myc-Erm, and Flag-PntP1 were co-expressed, we could also pull down all three proteins in the same complex (S7F Fig, lanes #2, #6, #10, and #14), indicating that Six4, Erm, and PntP1 form a complex. Since there is no detectable expression of endogenous PntP1 in S2 cells (S7A and S7B Fig), the interaction between Six4 and Erm is likely direct, but we do not completely rule out the possibility that other unknown proteins mediate their interaction.

Next, we tested whether co-existence of Six4, Erm, and PntP1 would make the complex more stable than a complex that contains only two of them. To this end, we first performed co-IP for a combination of any two of these three proteins with graded concentrations (150mM to 1200mM) of NaCl. Our results showed that with the increase of the salt concentration, the co-IP became less efficient. However, even at 1200mM of NaCl, we did not see complete disruption of the interactions between any two of these proteins (S7C–S7E Fig). Then we compared the co-IP efficiency between the complex that contains all these three proteins and a complex that contains only two of them with 150mM or 1200mM of salt. We found that with 1200mM of salt, co-expression of Flag-PntP1 significantly enhanced the binding between Six4-HA and Myc-Erm. The % co-IPed Myc-Erm with the anti-HA antibody was increased from 6.7% to 13.6% in the presence of Flag-PntP1 (S7F Fig (lanes #6 and #8) and S7F' Fig). However, with 150mM of NaCl, no significant enhancement was observed probably because at the low concentration of salt, the binding between Myc-Erm and Six-HA was more stable. In addition to the enhancement of the binding between Myc-Erm and Six4-HA by the presence of Flag-PntP1, we found that the presence of Six4-HA probably also enhanced the binding between Flag-PntP1 and Myc-Erm. The % co-IPed Myc-Erm with the anti-Flag antibody was increased by about 35% with 150mM of NaCl and 50% with 12000mM of NaCl when Six4-HA was co-expressed (S7F Fig (lanes #9, #10, #13, and #14) and S7F" Fig). Although the enhancement was not statistically significant when the statistical analyses were done separately for the co-IPs carried out with 150mM and 1200mM of NaCl probably because of the small sample size (n = 3 for each condition), we did see statistically significant difference of the % co-IPed Myc-Erm when all the data from two different salt concentrations were pooled together for a paired t-test (S7F''' Fig).

Taking together, all these co-IP data suggest that 1) Six4, Erm, and PntP1 can form a trimeric complex; and 2) the presence of all these three proteins makes the complex more stable, which may in turn confers stronger inhibition of PntP1's activity.

## Six4 acts together with PntP1 in newly generated imINPs to prevent premature differentiation of INPs

Our results demonstrate that Six4 inhibits PntP1's activity and expression during the maturation of INPs. However, based on the expression of Six4-GFP, Six4 is also expressed in type II

NBs and newly generated imINPs. Our gain-of-function analyses showed that when Six4 was overexpressed in these cells using *pntP1-GAL4* as a driver, it inhibits the expression and function of PntP1. What does Six4 do in these cells at its physiological expression levels? Does it still inhibit PntP1's activity and expression or have other functions? To answer this question, we tried to determine if PntP1's activity or expression would be enhanced in these cells when Six4 was knocked down. However, it might be difficult to assess the enhancement of PntP1 activity in these cells because overexpression of PntP1 in type II NB lineages does not produce any obvious phenotypes (Fig 5F) [25]. Therefore, we tried to perform double knockdown of Six4 and PntP1 in these cells using *pntP1-GAL4* as a driver. The rationale is that if Six4 inhibits PntP1's expression and/or activity, simultaneous knockdown of Six4 might be able to restore the activity/expression of PntP1 to some extent and partially rescue the PntP1 knockdown phenotypes in these cells, such as loss of mINPs and/or ectopic Ase expression in the NB. Our previous studies have shown that PntP1 inhibits Ase expression in type II NBs and Pros expression in the newly generated imINPs (Fig 1B) [24,29]. Knockdown of PntP1 led to ectopic Ase activation in about 80% of type II NBs and depletion of mINP in over 90% of type II NB lineages. In addition, the total number of type II NBs was also increased by 2–3 folds due to dedifferentiation of imINPs (Fig 7A–7A$_1$ and 7C–7E) [29]. However, in contrast to what we expected, we found that the percentage of type II NBs with ectopic Ase expression in Six4 and PntP1 double knockdown type II NBs remained the same as those in PntP1 knockdown type II NBs (Fig 7B and 7C), indicating that PntP1's activity or expression was not enhanced in PntP1 and Six4 double knockdown type II NBs. Even more surprisingly, the loss of mINPs phenotype was enhanced rather than suppressed by simultaneous Six4 knockdown, resulting in complete depletion of mINPs in all Six4 and PntP1 double knockdown type II NB lineages. Instead, only a few Ase⁺ GMCs were observed next to the NB (Fig 7B–7D). Furthermore, simultaneous knockdown of Six4 fully inhibited the supernumerary type II NB phenotype resulting from PntP1 knockdown. There were only 8 type II NBs per brain lobe in Six4 and PntP1 double knockdown larvae (Fig 7A, 7B and 7E). The suppression of the supernumerary type II NB phenotype in Six4 and PntP1 double knockdown brains was unlikely due to transformation of type II NBs into Ase⁺ type I-like NBs because 20% of the double knockdown NBs remained Ase⁻ (Fig 7C) and total number of Ase⁺ GFP⁻ type I NBs was not increased (Fig 7E). Similar results were also observed when Six4 was knocked down in *pntP1⁹⁰* mutants (Fig 7F–7J). *pntP1⁹⁰* is a *pntP1*-specific mutant allele carrying a small indel in a *pntP1*-specific exon and produces a truncated form of PntP1 that only contains the N-terminal 180aa [29]. In *pntP1⁹⁰* mutant larvae, there were about 20 Ase⁻ type II NBs per brain lobe and the number of mINPs was also reduced to about 5 per type II NB (Fig 7G–7G₁, 7I and 7J). When Six4 was knocked down in *pntP1⁹⁰* mutants, there were only 8 Ase⁻ type II NBs per brain lobe and mINPs were also completely eliminated (Fig 7H–7J). These results indicate that knockdown of Six4 enhances the loss of mINPs and suppresses the generation of supernumerary type II NBs resulting from the loss of PntP1.

Our previous studies have demonstrated that ectopic nuclear Pros expression in newly generated imINPs accounts for the premature differentiation of INPs into GMCs and subsequent loss of mINPs resulting from the loss of Btd or PntP1 [29,30], we thus investigated if the enhanced loss of mINPs in Six4 and PntP1 double knockdown brains was because of enhanced ectopic expression of nuclear Pros in imINPs. We examined the expression of Pros in newly generated imINPs, which were identified as Ase⁻ cells right next to the type II NBs, at different developmental stages. In PntP1 knockdown brains, about 2–5 newly generated imINPs pre brain lobe remained Pros⁻ until 84 hrs after egg lying (AEL) (Fig 7K–7K" and 7O) although nearly all newly generated imINPs became nuclear Pros⁺ at around 100h AEL (Fig 7L–7L" and 7O). However, in Six4 and PntP1 double knockdown brains, essentially all newly generated

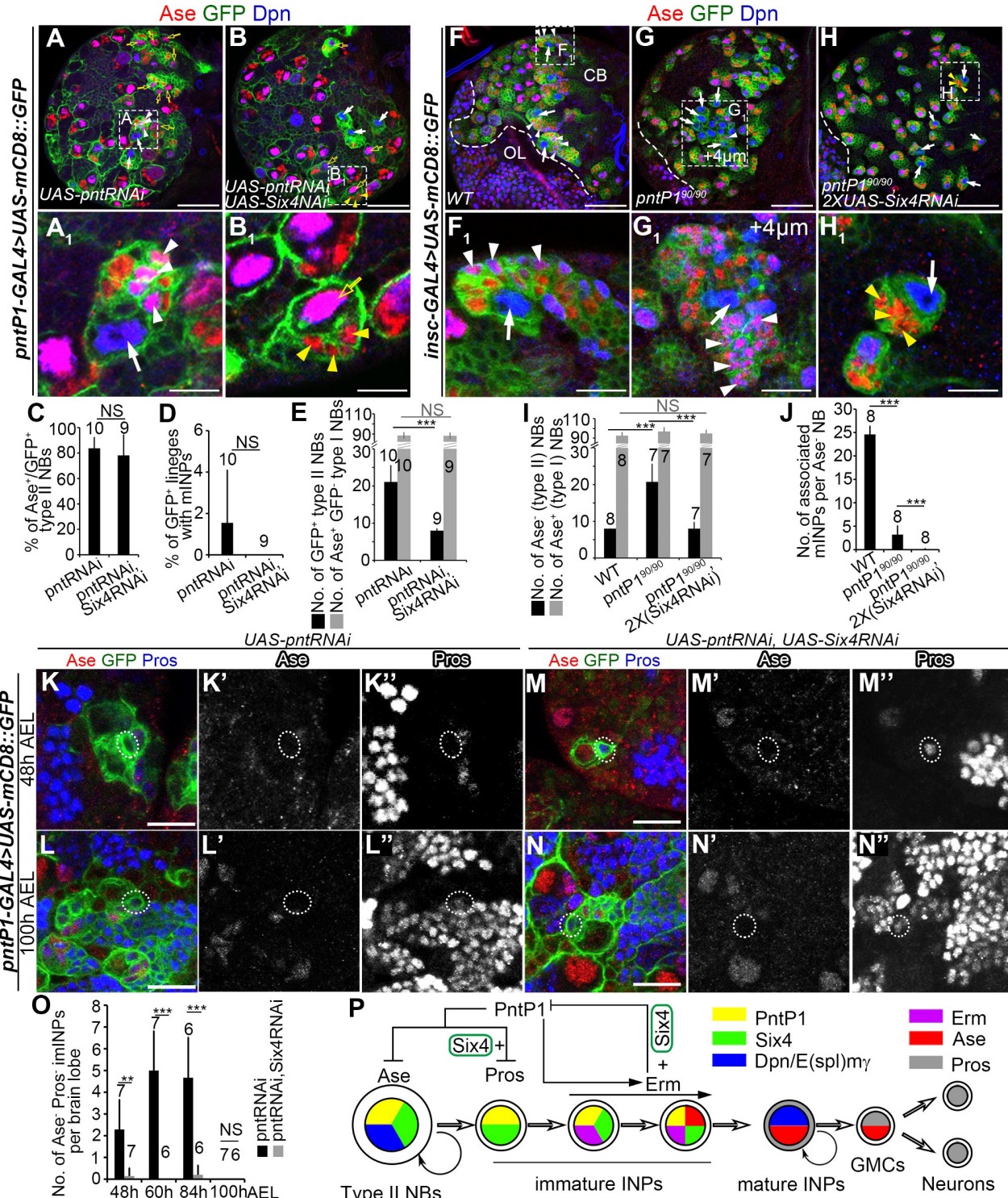

**Fig 7. Six4 acts in newly generated imINPs to prevent premature differentiation of INPs.** Type II NB lineages are labeled by mCD8-GFP driven by *pntP1-GAL4* (A-B₁, K-N") or *insc-GAL4* (F-H₁). Brains are counterstained with antibodies against Dpn, Ase and Pros. Scale bars equal 50μm in (A, B, F, G, H) or 10μm in (A₁, B₁, F₁, G₁, H₁ and K-N"). (A-B₁) An increased number of type II NBs is observed in PntP1 knockdown brain (A-A₁) but not in a PntP1 and Six4 double knockdown brain (B-B₁). However, ectopic Ase expression in type II NBs (open yellow arrows) is similarly observed in PntP1 knockdown (A-A₁) and PntP1 and Six4 double knockdown (B-B₁) brains. White arrows point to Ase⁻ type II NBs. White arrowheads point to mINPs and yellow arrowheads to GMCs. (A₁) and (B₁) are enlarged views of the lineages highlighted with dashed squares in (A) and (B), respectively. (C-E) Quantifications of the percentage of Ase⁺ type II NBs (C), the percentage of type II NB lineages with mINPs (D), and

total number of type II NBs or Ase$^+$ GFP$^-$ type I NBs (E) in PntP1 knockdown or PntP1 and Six4 double knockdown brains. The number on top of each bar represents the number of brains examined. $^{***}$, $p < 0.001$; NS, not significant. (F-H$_1$) The number of type II NBs is increased in $pntP1^{90}$ mutant brains (G) but remains the same in $pntP1^{90}$ mutant brains with the expression of $UAS$-$Six4$ $RNAi$ driven by $insc$-$GAL4$ (H) as in the wild type (F). White arrows point to type II NBs and white arrowheads to mINPs. (F$_1$, G$_1$, and H$_1$) are enlarged views of the areas highlighted with dashed squares in (F, G, and H), respectively, to show that mINPs are generated in the wild type (F$_1$) and $pntP1^{90}$ mutant brains (G$_1$) but not in the $pntP1^{90}$ mutant brains with the expression of $UAS$-$Six4$ $RNAi$ (H$_1$), which produce GMCs (yellow arrowheads) directly instead. (I-J) Quantifications of the number of Ase$^-$ type II NBs or Ase$^+$ type I NBs/brain lobe (I) and number of mINPs per Ase$^-$ NB (J). The number on top of each bar represents the number of brains examined. $^{***}$, $p < 0.001$; NS, not significant. (K-L") Ectopic nuclear Pros expression is not observed in the newly generated imINP (dotted circles) in a PntP1 knockdown type II NB lineage (K-K") at 48hrs AEL but is detected in all newly generated imINPs at 100 hrs AEL (L-L"). (M-N") Ectopic nuclear Pros is detected in all newly generated imINPs (dotted circles) in PntP1 and Six4 double knockdown type II NB lineage at 48 hrs (M-M") and 100 hrs (N-N") AEL. (O) Quantifications of the number of newly generated imINPs without the ectopic nuclear Pros expression in PntP1 knockdown and PntP1 and Six4 double knockdown brains. The number on top of each bar represents the number of brains examined. $^{**}$, $p < 0.01$; $^{***}$, $p < 0.001$; NS, not significant. (P) A schematic diagram shows potential functions of Six4 in type II NB lineage development. Late during imINP development, Six4 acts together with Erm to inhibit PntP1's activity and expression, which ensures that imINPs become fate-committed mINPs instead of dedifferentiating into type II NBs. In the newly generated imINPs, Six4 also contributes to the inhibition of nuclear Pros expression, thus preventing premature differentiation of INPs into GMCs.

imINPs had already turned on the expression of nuclear Pros at 48 hrs AEL (Fig 7M–7O) although knockdown of Six4 alone did not lead to ectopic Pros expression in newly generated imINP (S8 Fig), indicating that the ectopic Pros expression was enhanced in Six4 and PntP1 double knockdown brains. The enhanced ectopic Pros expression in all newly generated imINPs likely accounts for the premature differentiation of INPs into GMCs and complete depletion of mINPs in all Six4 and PntP1 double knockdown type II NB lineages. It probably also accounts for the suppression of the supernumerary type II NB phenotype because the premature differentiation of INPs into GMCs and subsequent cell cycle exit resulting from the enhanced ectopic Pros expression in all newly generated imINPs (Fig 7O) makes it impossible for imINPs to dedifferentiate back to the type II NBs as demonstrated in $btd$ $brat$ double mutant type II NB lineages [31]. Taken together, these results suggest that at its physiological expression level, Six4 may also contribute to the suppression of Pros in newly generated imINPs as PntP1 and Btd do.

## Discussion

Avoiding both dedifferentiation and premature differentiation is critical for maintaining homeostasis of INPs in developing brains. In this study, we identified a new transcription factor, Six4, that is expressed in type II NB lineages to prevent both dedifferentiation and premature differentiation of INPs. Six4 prevents dedifferentiation of imINPs by acting together with Erm to inhibit PntP1's activity and expression in late during imINP maturation. In the newly generated imINPs, Six4 also contributes to the inhibition of Pros expression to prevent premature differentiation of INPs into GMCs (Fig 7P). Our work not only identifies a novel transcription factor that regulates the development of INPs, but also provides molecular insights into the mechanisms underlying the inhibition of PntP1 activity by Erm and Six4.

### Six4 prevents dedifferentiation of imINPs

The main phenotype resulting from Six4 knockdown in type II NB lineages is the generation of supernumerary type II NBs. We provide several lines of evidence to demonstrate that the generation of supernumerary type II NBs is likely due to dedifferentiation of imINPs. First, asymmetric division of type II NBs is not affected by Six4 knockdown as demonstrated by the normal asymmetric segregation of aPKC and Mira. Second, the expression pattern of Six4-GFP indicates that Six4 could function in imINPs. Third, the generation of supernumerary type II NBs resulting from the knockdown of Six4 in imINPs, although the phenotype is weak due to the persistence of Six4 expression in imINPs, provides evidence to support that

Six4 could functions in imINPs to prevent dedifferentiation of imINPs. The persistence of Six4 proteins after expressing *UAS-Six4 RNAi* in imINPs could be due to the following two reasons. One is that knockdown of Six4 in imINPs with *erm-GAL4* may not be very efficient. The imINP state only lasts about 8–10 hrs [23]. *erm-GAL4* expression in imINPs is not turned on immediately and does not reach its peak at least until 2–3 hrs after the birth of imINPs [23]. There may not be enough time for significant knockdown of Six4 because of the short duration of the imINP state and the delayed expression of *erm-GAL4* in imINPs. Another reason could be that Six4 proteins expressed in type II NBs may be inherited by imINPs and continue to function in imINPs if these proteins cannot be degraded soon enough. Fourth, the genetic and biochemical interactions between Six4 and Erm suggest that Six4 functions in the same pathway as Erm, which has been well demonstrated for its role in preventing dedifferentiation of imINPs [26,41,49]. Finally, the rescue of the supernumerary type II NB phenotype resulting from Erm knockdown by overexpressing Six4 specifically in imINPs also supports that Six4 could functions in imINPs to prevent their dedifferentiation. However, to provide definitive evidence to demonstrate that that the supernumerary type II NBs resulting from Six4 knockdown are generated from dedifferentiation of imINPs, it will be essential to perform a lineage tracing experiment to follow the fate of the progeny of imINPs. Unfortunately, we were not able to perform the lineage tracing experiment due to some technical difficulties.

Our results suggest the majority of supernumerary type II NBs resulting from Six4 knockdown are probably generated at embryonic stages and possible also early larval stages. First, knocking down Six4 specifically at larval stages with *pntP1-GAL4* together with *tub-GAL80$^{ts}$* results in much fewer supernumerary type II NBs than knocking down Six4 as soon as type II NBs are specified at embryonic stage with *pntP1-GAL4* alone. Second, most of the supernumerary type II NBs established their own independent lineages and coexistence of multiple type II NBs, which are presumably generated from dedifferentiation of imINPs, is only observed in a small percentage of single isolated type II NB lineages at late larval stage. Although the exact underlying mechanisms remain to be investigated, the supernumerary type II NBs seems unlikely to be generated by aberrant specification of NBs from the neuroectoderm because the *pntP1-GAL4* driver used for Six4 knockdown is only expressed in type II NB lineages but not in type I NBs or the neuroectoderm before type II NBs are specified [24,45,46]. A recent study has shown that GMCs generated from type I NBs and INPs from type II NBs during early development are more susceptible to malignant transformation because of high expression of the BTB-zinc finger protein Chinmo [51]. Chinmo is expressed in young NBs and their progeny and its expression gradually decreases with the progression of the development. After puparium formation, Chinmo expression is suppressed. Chinmo was first identified as a neuronal temporal identity gene and is generally involved in specification of larval-born neurons, including those generated in type II NB lineages [51–54]. Later studies suggest that Chinmo also promotes the self-renewal of stem cells and tumor growth [55,56]. In the absence of Chinmo, the growth of GMC- or INP-derived tumors resulting from the loss of Pros or Brat, respectively, is significantly suppressed [51]. Therefore, one interesting possibility could be that the majority of the supernumerary type II NBs resulting from Six4 knockdown are generated early during development when Chinmo expression is high. At embryonic stages, type II NBs also divide to produce imNPs [45,46], could dedifferentiate to become type II NBs when Six4 is knocked down. When Six4 is knocked down at later developmental stages, the chance of imINPs to initiate tumorigenic dedifferentiation might be much lower because of significantly reduced expression of Chinmo. It would be interesting to investigate in the future how manipulating Chinmo expression might affect the generation of supernumerary type II NBs resulting from Six4 knockdown.

Previous studies demonstrate that one main driving force for the dedifferentiation of imINPs is PntP1 [22,23]. Other factors, such as Brat, Numb, and Erm, that are required for preventing dedifferentiation of imINPs, all function by terminating PntP1's function/expression in imINPs either directly or indirectly [18,19,22,23,25]. Our results suggest that Six4 prevents dedifferentiation of imINPs similarly by inhibiting PntP1's function and expression in imINPs. From our loss-of-function studies, we show that knockdown of Six4 leads to delayed termination of PntP1 expression, which results in ectopic expression of PntP1 in mINPs. From our gain-of-function analyses in both type I and type II NB lineages, we demonstrate that Six4 can inhibit PntP1's function. In type I NB lineages, misexpressing Six4 blocks the PntP1 misexpression-induced suppression of Ase and generation of INP-like cells. In type II NB lineages, overexpressing Six4 inhibits the activation of Erm by PntP1 even if PntP1 is overexpressed. Results from our biochemical studies suggest that Six4 inhibits PntP1's activity probably by forming a complex with PntP1. Six4 may function as a transcriptional repressor by recruiting co-repressors such as Groucho like in other tissues [36,57]. By recruiting co-repressors, Six4 may inhibit PntP1 transcriptional activity. Alternatively, binding of Six4 may disrupt binding of PntP1 to DNAs or its tissue-specific co-factor(s). However, it is not clear whether the suppression of PntP1 expression by Six4 has anything to do with the inhibition of PntP1's activity or involves a totally independent mechanism. One possibility could be that PntP1 activates its own expression for maintaining the expression and inhibition of PntP1's activity by Six4 disrupts the positive self-regulatory loop of PntP1, which eventually leads to termination of PntP1 expression in imINPs and maturation of INPs. Such positive self-regulation is a common mechanism in regulating gene expression [58]. In support of this notion, inhibition of PntP1's activity seems to occur before the termination of PntP1 expression in imINPs because PntP1 coexists with Ase in imINPs late during imINP development, which indicates PntP1 is no longer functioning to suppress Ase in these cells before PntP1 expression is lost. In order to elucidate exactly how Six4 inhibits PntP1's activity and expression, it will be essential to examine the interactions between Six4 and PntP1 specifically in type II NB lineages and map the DNA binding profiles of Six4 and PntP1 in the future.

## Functional relationship between Six4 and Erm in preventing dedifferentiation of imINPs

Findings from this study suggest that Six4 and Erm function similarly to prevent dedifferentiation of imINPs by antagonizing the activity and expression of PntP1. Six4 and Erm could act in the same pathway or in two independent pathways in parallel to perform their functions. We rule out the possibility of cross-regulation between Six4 and Erm by showing that knocking down the expression of either one of them does not affect the expression of the other. However, Six4 and Erm genetically interact in preventing dedifferentiation of imINPs. We show that the supernumerary type II NBs phenotype resulting from Six4 knockdown is significantly enhanced in $erm^2$/+ heterozygous mutant background and that double knockdown of Six4 and Erm leads to synergistic enhancement of the supernumerary type II NB phenotype. The genetic interaction suggests that Six4 and Erm function in the same pathway even though they do not regulate each other's expression. In support of these genetic interaction data, our biochemical studies reveal that Six4, Erm, and PntP1 could form a complex. Therefore, Six4 and Erm likely act together in the same complex to inhibit PntP1 activity in imINPs and both Six4 and Erm are required for complete inhibition of PntP1 activity. The presence of all three proteins, Six4, Erm, and PntP1, is probably essential for the stability of the complex as our co-IP data demonstrate that the presence of Flag-PntP1 or Six4-HA enhances the binding between Myc-Erm and Six4 or between PntP1 and Erm, respectively. In the absence of Six4, PntP1 and

Erm may form a less stable complex and some PntP1 proteins may remain free and active, which will lead to activation of its target genes and subsequent dedifferentiation of imINPs. In addition to increasing the stability of the complex, the presence of both Six4 and Erm in the complex may also help recruit different co-factors that are essential for complete inhibition of PntP1's activity in imINPs. For example, Six4 may recruit co-repressor Groucho that will confer transcriptional repression activity, whereas Erm may recruit epigenetic modifiers SWI/SNF complex proteins and/or HDAC proteins to make the genetic loci that contain PntP1 target genes inactive. However, it is unlikely that the presence of both Six4 and Erm is just for recruiting different co-factors because it will be difficult to explain why overexpressing Six4 can rescue the Erm knockdown phenotype if all PntP1 proteins have already bound to Six4 in the absence of Erm. Excessive amounts of Six4 proteins would not help to recruit the co-factor(s) that are normally only recruited by Erm to the complex because Six4 and Erm belong to different protein families and do not share protein domains or structural similarities.

Our working model that a complete inhibition of PntP1's function requires its formation of a complex with both Six4 and Erm probably also explains why PntP1 activity is normally not inhibited by Six4 in type II NBs and newly generated imINPs even though Six4 is expressed in these cells. It could be simply because Erm is not expressed in these cells. Therefore, Six4 can only inhibit PntP1 activity in type II NBs when it is overexpressed and even overexpressed Six4 can only partially inhibit PntP1's function. In contrast, when Erm is misexpressed in type II NBs, it can fully inhibit PntP1's function as indicated by the ectopic Ase expression and the loss of INPs in all type II NB lineages [25,41] because endogenous Six4 is already expressed in type II NBs.

## A potential role of Six4 in preventing premature differentiation of INPs

In addition to preventing dedifferentiation of imINPs by inhibiting the function and expression of PntP1, our results suggest that Six4 might also be involved in preventing premature differentiation of INPs. Although knockdown of Six4 alone in type II NB lineages does not lead to dramatic loss of mINPs, knockdown of Six4 completely eliminates mINPs in PntP1 knockdown or mutant type II NB lineages. The elimination of mINPs is not because of complete transformation of type II NBs into type I NBs because knockdown of Six4 does not affect the expression of Ase in the PntP1 knockdown or mutant type II NBs. Rather, it was because of enhanced ectopic expression of Pros in newly generated imINPs, which leads to premature differentiation of INPs into GMCs. Our results show that in Six4 and PntP1 double knockdown brains, nuclear Pros is ectopically expressed in all newly generated imINPs at 48 hrs AEL, whereas in PntP1 knockdown brains, 2–5 newly generated imINPs in each brain lobe remain Pros-negative at the same stage. Although we did not examine the expression of Pros before 48 hrs AEL in Six4 and PntP1 double knockdown larvae, the ectopic nuclear Pros is probably expressed in every single newly generated imINPs at earlier developmental stages as soon as they are produced from type II NBs because the generation of supernumerary type II NBs is completely inhibited in these animals. The ectopic expression of nuclear Pros inhibits the dedifferentiation of imINPs as demonstrated in *btd brat* double mutant type II NB lineages [31]. Thus, the significant enhancement of the ectopic Pros expression suggests that Six4 likely acts together with PntP1 to suppress Pros expression in newly generated imINP and prevent premature differentiation of INPs. This is quite surprising considering that late during imINP development Six4 acts together with Erm to inhibit PntP1's expression and function. The opposite effects of Six4 on PntP1's function at different stages of imINP development indicate that Six4's function might be context-dependent. It could be due to availability of different co-factor(s) that bind to Six4, such as Erm or other factors, or other unknown mechanisms. Six4

and PntP1 might have partial redundancy in suppressing Pros expression in newly generated imINPs, but PntP1 could have a more prominent role. Alternatively, Six4 may not be involved in the suppression of Pros in normal type II NB lineages, but only when PntP1's expression is reduced and/or the development of type II NB lineages is impaired would Six4 somehow become a suppressor of Pros expression. Our previous studies have shown that Btd is also required to suppress Pros expression in newly generated imINPs and prevent premature differentiation of INPs into GMCs [30]. It would be interesting to test the functional relationship between Six4 and Btd in suppressing Pros expression in newly generated imINPs.

## Materials and methods

### Fly stocks

The *Six4-GFP* line (#67733, Bloomington *Drosophila* Stock Center [BDSC], Bloomington, Indiana) was used to detect Six4 expression. *E(spl)mγ-GFP* reporter line [59] was used to label NBs and mINPs. GAL4 lines, including *pntP1-GAL4* (or *GAL4^{14-94}*) [24], *erm-GAL4(II)*, *erm-GAL4(III)* [48,49] and *insc-GAL4* [60], were used for UAS transgene expression. *UAS-Six4 RNAi^{HM05254}* (#30510, BDSC) and three other additional independent *UAS-Six4 RNAi* lines (#30456, #48598, and #104958 from Vienna *Drosophila* Resource Center [VDRC, Vienna, Austria]) were used for knocking down Six4. Knockdown of Erm was carried out with *UAS-Erm RNAi* (#26778, BDSC). Knockdown of *pnt* was carried out with *UAS-pnt RNAi* (#35038, BDSC). FlyORF #F000049 line (named *UAS-Six4ORF-HA or UAS-Six4ORF* in this study) was used for mis-/over-expression of Six4. *UAS-PntP1* [24] was used for mis-/over-expression of PntP1. Other fly lines used in this study include *pros^{17}* [61], *erm^2* [41], and *pntP1^{90}* [29]. *Tub-GAL80^{ts}* line (#7019, BDSC) were used for temporal control of *UAS-Six4 RNAi* expression.

### UAS-transgene expression

For *RNAi* knockdown, except in Fig 6D–6G, embryos were collected for 8~10 hours at 25˚C and shifted to 29˚C to boost the efficiency. In Fig 6D–6G, embryos were collected for 8~10 hours and raised at 25˚C. *UAS-dcr2* was coexpressed with *UAS-RNAi* transgenes to enhance the RNAi knockdown efficiency. For mis-/over-expression of Six4 or PntP1, embryos were collected for 8~10 hours at 25˚C and shifted to 29˚C. When *tub-GAL80^{ts}* was used to temporally control the expression of *UAS-Six4 RNAi*, embryos were collected for 8~10 hours and raised at 18˚C for 2 days. Then hatched larvae were cultured at 29 ˚C for additional 3–4 days before dissection.

### Immunostaining and confocal microscopy

2^{nd} or 3^{rd} instar larvae were sacrificed and dissected. Brain lobes along with VNCs were immunostained as previously described [62] except that the fixation duration was increased to 35 mins. Immunocytochemistry for transfected S2 cells at log phase were performed as described previously [63]. Primary antibodies used are chicken anti-GFP (Catalog #GFP-1020, Aves Labs, Tigard, Oregon; 1:500–1000), rabbit anti-DesRed (Catalog #632392, Takara Bio USA, Inc., Mountain View, California; 1:250), rabbit anti-Dpn (1:500) (a gift from Dr. Y.N. Jan) [64], guinea pig anti-Ase (a gift from Dr. Y.N. Jan, 1:5000) [13], rat anti-Erm (a gift from Dr. C. Desplan; 1:300), rabbit anti-PntP1 (a gift of J. B. Skeath; 1:500) [65], mouse anti-Pros (Developmental Studies Hybridoma Bank; 1:20), rabbit anti-Miranda (a gift from Dr. Y.N. Jan, 1:1000), mouse anti-aPKC (Catalog #sc-17781, Santa Cruz Biotechnology, Inc., Dalla, Texas, 1:500) and mouse anti-Flag antibody (Catalog #F1804, Sigma-Aldrich Co, St. Louis,

Missouri; 1:500). Secondary antibodies conjugated to Daylight 405 (1:300–400), Daylight 488 (1:100), Cy3 (1:500), Rhodamine Red-X (1:500), Daylight 647 (1:500) or Cy5 (1:500) used for immunostaining are from Jackson ImmunoResearch (West Grove, Pennsylvania). Images were collected using a Carl Zeiss LSM780 confocal microscopy and processed with Adobe Photoshop. Student's T-test or Welch's T-Test, or Mann-Whitney test (for data not normally distributed) was used for comparing the number of type II NBs or INPs. The data used to generate all the related graphs and statistical analyses can be found in S1 Data.

## Plasmid construction and generation of transgenic lines

The *UAS-Six4$^{HM05254Resis}$* construct was generated by assembling the following three overlapping DNA fragments that covers the entire transcript of the *six4* gene with the Gibson Assembly Cloning technology: a middle fragment of 391 bps that covers the 3'-end of exon 2, intron 2, and the 5'-end of exon 3; a 5'-end fragment that includes a sequence from exon 1, intron 1, and 5'-part of exon 2 with a 40-bp overlap with the middle fragment; and a 3'-end fragment that include a sequence from 3'-part of exon 3 to the last exon (exon 5) with a 45-bp overlap with the middle fragment. The middle fragment contains the *dsRNA-HM05254* target sequence [66] and was synthesized with the introduction of silent mutations to the *dsRNA-HM05254* target sequence according to the optimal *Drosophila* codon. The sequence of the middle fragment is shown below (The underlined parts are the regions that are targeted by *dsRNA-HM05254* and carry silent mutations indicated by lowercase. The sequence between the two underlined parts is intron 2 of *six4*).

5'gaagtggcggcggattgggggggcaatgccggcagtggtggccacc<u>TgATcagcAAcCTgACgGCtGCgCAtAAt ATGagcGCtGTctcctccTTcCCcATtGAcGCgAAaATGCTcCAaTTtTCgACcGAcCAG</u>gtaggata cttctcccatatatcctagacttttgcagacataaagtgctatgaacttcttctttag<u>ATtCAaTGtATGTGtGAaGCcCTGC AaCAaAAaGGAGAtATtGAaAAaCTGACgGACcTTtCTgTGCtccCTGCCcCCgtccGAaTTtTT CAAaACaAAtGAatccGTgCTGCGtGCtCGtGCgATGGTcGCCTAtAAcCTcGGaCAaTTtCA CGAaCTcTACAAt</u>ctgctggagacgcactgcttttcgat3'.

The 5'- and 3'-end fragments were amplified with CloneAmp HiFi PCR Premix (Takara Bio., Mountain View, California) from genomic DNAs using primers shown below. The underlined parts of these primers are the sequences that overlap either with the sequence of the middle fragment (primers Six4J1R and Six4J3F) or with the sequence of the pJET vector (primers Six4J1F and Six4J3R).

Six4J1F:5'- <u>agctgagaatattgtaggagatcttctagaaagatttcgctggc</u>ttgcagtcgagtttga-3'
Six4J1R: 5'-<u>ccaccactgccggcattgcccccccaatccgccgccacttc</u>tctgcaggccggcgcataga-3'
Six4J3F: 5'-<u>atttcacgaactctacaatctgctggagacgcactgctttttcgat</u>caagtaccacgtgga-3'
Six4J3R: 5'-<u>agatcttccggatggctcgagtttttcagcaagatgtttgctggccagctcatcgaagaa</u>-3'

The above three overlapping fragments together with a commercially linearized pJET1.2/ blunt vector (Thermo Fisher Scientific, Waltham, Massachusetts) were assembled through Gibson assembly (Catalog # E2621S, New England Biolabs, Ipswich, Massachusetts). The assembled *dsRNA-HM05254-resistant Six4 gene (named as Six4$^{HM05254Resis}$* was excised with BglII/XhoI from the pJET vector and subcloned into pUAST. The construct was injected into *Drosophila* embryos by the Rainbow Transgenic Flies, Inc. (Camarillo, California) and the transformants were generated based on a standard P-element transposon protocol.

For generating the pUAST-Six4-3×HA construct, the *Six4-3×HA* fusion gene was amplified from FlyORF #F000049 line with CloneAmp HiFi PCR Premix (Takara Bio USA, Inc.) and inserted into pUAST vector between BglII/XhoI sites. For generating the pAMW-EGFP construct, the EGFP sequence was amplified from pH-stinger vector (*Drosophila* Genomics Resource Center, Bloomington, Indiana) and was used to replace the *erm* fragment between

AgeI/NheI sites in pAMW-Erm. pAW empty vector was generated by deleting *Flag-PntP1* cassette from pAFW-PntP1. Primers used for making pUAST-Six4-3xHA and pAMW-EGFP constructs are listed below.

Six4f: 5'- AAATAGATCTATGTTTGACAAGAATTTGGACGGCAA -3'
Six4-HAr: 5'- AAATCTCGAGACGCTTAGTGCTAGCGTCAA -3'
EGFPf: 5'-AAATACCGGTATATGGTGAGCAAGGGCGAGGA -3'
EGFPr: 5'-TTAAGCTAGCTTACTTGTACAGCTCGTCCATGCCGAGA -3'

## S2 cell culture, transfection, and co-immunoprecipitation

*Drosophila* S2 cells were maintained and transfected in Schneider's Medium (Cataglog #21720004, Thermo Fisher Scientific) containing 10% Heat-Inactivated Fetal Bovine Serum (Catalog #10082147, Thermo Fisher Scientific) following *Drosophila* Schneider 2 (S2) Cells USER GUIDE (Thermo Fisher Scientific). pAFW-PntP1, pAMW-Erm (both are gifts from Dr. H. Y. Wang) [26], and pAMW-EGFP constructs were used for expressing Flag-PntP1, Myc-Erm, and Myc-EGFP in S2 cells under the control of the *Act5C* promoter. pUAST-Six4-3xHA and copper inducible pMT-GAL4 [67] constructs were co-transfected for the expression Six4-HA. Mock vectors pUAST and pAW were co-transfected as controls to balance the amounts of each vector transfected. 5μg of each vector were co-transfected by using Calcium Phosphate Transfection Kit (Catalog #K278001, Thermo Fisher Scientific). 500 μM $CuSO_4$ (Catalog #451657, Sigma-Aldrich Co., St. Louis, Missouri) was used to induce GAL4 expression 1 day after transfection.

For co-immunoprecipitation (co-IP), S2 cells were collected 72 hrs after $CuSO_4$ induction for protein homogenization in 1% NP-40 lysis buffer (50 mM Tris-Cl, 1% NP-40, pH 8.0, with 150mM NaCl unless the salt concentration is particularly indicated as in S7 Fig) with protease inhibitor cocktail (Catalog #78430, Thermo Fisher Scientific). Immunoprecipitation was performed by overnight incubation at 4˚C with 2μg rabbit anti-HA mAb (Catalog #MA5-27915, Thermo Fisher Scientific), mouse anti-Myc mAb (Catalog #MA1-21316, Thermo Fisher Scientific), or mouse anti-Flag antibody (Catalog #F1804, Sigma-Aldrich Co, St. Louis, Missouri), followed by incubation with protein A/G agarose beads (Catalog #sc-2003, Santa Cruz Biotechnology, Inc.) (pre-blocked with 5% BSA) at 4˚C for 4~5 hrs. Collected beads were washed 4~5 times with cold lysis buffer containing protease inhibitor cocktail and eluted for SDS-PAGE separation. Proteins transferred to PVDF were blotted with mouse anti-HA antibody (Catalog #sc-7392, Santa Cruz Biotechnology Inc.), rabbit anti-HA mAb (Catalog #MA5-27915, Thermo Fisher Scientific), rabbit anti-Myc antibody (Catalog #2278S, Cell Signaling Technology, Danvers, Massachusetts), or mouse anti-Flag antibody (Catalog #F1804, Sigma-Aldrich Co, St. Louis, Missouri) or rabbit anti-PntP1 (a gift of J. B. Skeath; 1:500). Secondary antibodies used were HRP conjugated anti-mouse IgG (Catalog #7076S, Cell Signaling Technology) or anti-rabbit IgG (Catalog #7074S, Cell Signaling Technology). Chemiluminescent assay was performed with chemiluminescent substrate (Catalog #34577 or #34095, Thermo Fisher Scientific). Images were collected with ChemiDoc Imaging Systems (Bio-Rad, Hercules, California).

For quantifying the co-IP, the intensities of the related bands from the western blot were first determined by Image Lab Software (Bio-Rad, Hercules, California). The % co-IP was calculated by dividing the co-IPed protein by the 100% input of the co-IPed protein, then normalized by the IPed protein to correct the differences in immunoprecipitation. Three replicates were performed for each condition and the paired t test was used for statistical analyses. The data used to generate all the related graphs and statistical analyses can be found in S1 Data.

## Supporting information

**S1 Fig. Knockdown of Six4 leads to the generation of supernumerary type II NBs.** Type II NB lineages are labeled with mCD8-RFP (in green) driven by *pntP1-GAL4* and counterstained with anti-GFP (in blue) and anti-Ase (in red) antibodies. Arrows point to type II NBs. Dashed lines demarcate the boundary between the central brain (CB) and the optic lobe (OL). Scale bars equal 50μm in (A-B, C-D, E-F and G) or 10μm in ($B_1$, $D_1$, $F_1$ and $F_2$). (A) The supernumerary type II NB phenotype resulting from the expression of *UAS-Six4 RNAi$^{HM05254}$* is not affected by expressing an additional copy of *UAS-mCD8-RFP*. (B-$D_1$) Expression of three additional independent *UAS-Six4 RNAi* transgenes results in consistent supernumerary type II NBs. ($B_1$) and ($D_1$) are enlarged views of the areas highlighted with dashed squares in (B) and (D), respectively, showing a single isolated lineage contains two type II NBs (arrows). Arrowheads in ($B_1$) and ($D_1$) point to mINPs. (E) A wild type larval brain lobe contains only 8 type II NBs. (F-G) Larval-specific expression of one (F-$F_2$) or two copies (G) of *UAS-Six4 RNAi* driven by *pntP1-GAL4* in combination with *tub-GAL80$^{ts}$* to larval stages still leads to the generation of extra type II NBs. ($F_1$) and ($F_2$) are enlarged views of the areas highlighted with dashed squares in (F). Note that two type II NBs (arrows) co-exist in a single lineage in both ($F_1$) and ($F_2$). (H) Quantifications of the number of type II NBs and mINPs in brains expressing independent *UAS-Six4 RNAi* transgenes. The number on top of each bar indicates the number of brains or lineages examined. ***, $p < 0.001$; NS, not significant. (I) Quantifications of the number of type II NBs in brains expressing one or two copies of *UAS-Six4 RNAi* after larval hatching. The number on top of each bar indicates the number of brains examined. ***, $p < 0.001$. (TIF)

**S2 Fig. Expression patterns of Six4-GFP in the *Drosophila* 3$^{rd}$ instar larval CNS.** 3$^{rd}$ instar larval brains and VNCs are stained with Phalloidin to outline individual NB lineages and counterstained with Dpn and Ase antibodies for identifying type I and type I NBs. Scale bars equal 50μm. (A-A') Six4-GFP is expressed in NBs and GMCs in two type I NB lineages in the VNC. Doted lines: the midline of the VNC. (B-C') Several type I NB lineages (white dashed lines) in ventral (B-B') and dorsal (C-C') brain lobes have Six4-GFP expression in NBs and GMCs. Yellow dashed lines in (C-C') outline type II NB lineages. (TIF)

**S3 Fig. Six4 is not involved in regulating the asymmetric cell division of type II NBs.** Type II NB lineages are labeled with mCD8-RFP driven by *pntP1-GAL4*. Mitotic marker pH3 staining is used to mark chromosomes. Scale bars equal 10μm. (A1-A2') In wild type type II NBs, Mira (red) is segregated to the basal cortex and aPKC (blue) is segregated to the apical cortex at the metaphase (A1-A1') and the anaphase/telophase (A2-A2'). The diagrams on the right show the distributions of Mira and aPKC at the metaphase and anaphase/telophase. (B1-B2') In Six4 knockdown type II NBs, Mira and aPKC are segregated to the basal and apical cortex, respectively, at the metaphase (B1-B1') and the anaphase/telophase (B2-B2') as in the wild type type II NBs. (TIF)

**S4 Fig. Expression of *UAS-Six4 RNAi* driven by *erm-GAL4* does not abolish Six4 expression.** (A-C) A larval brain expressing *UAS-Six4 RNAi* driven by *erm-GAL4(II)* together with *erm-GAL4(III)* (B) has the same number of type II NBs as a wild type brain does (A). (C) Quantifications of the number of type II NBs in the wild type brains and brains expressing UAS-six4 RNAi driven by *erm-GAL4 (II)* and *erm-GAL4(III)*. The number on top of each bar indicates the number of brains examined. NS, not significant. (D-E") Six4-GFP expression in wild type type II NB lineages (D-D") and type II NB lineages that express two copies of *UAS-*

*Six4 RNAi* driven by *erm-GAL4(II)* (E-E"). Open arrows point to Ase⁻ Dpn⁻ imINPs and yellow arrows point to Ase⁺ Dpn⁻ imINPs.
(TIF)

**S5 Fig. Six4 overexpression promotes ectopic nuclear Pros expression in imINPs and reduces PntP1 expression.** In all images, type II NB lineages are labeled with mCD8-GFP driven by *pntP1-GAL4* and counterstained with antibodies against Dpn, Ase, Pros and/or PntP1. Type II NB lineages are outlined by dashed lines. Scale bars equal 10μm. (A-A") In wild type type II NB lineages (dashed lines), nuclear Pros is not detected in either Ase⁻ (open white arrows) or Ase⁺ (yellow arrows) imINPs. White arrows point to type II NBs. (B-B") Nuclear Pros is ectopically expressed in Ase⁻ (open white arrows) and Ase⁺ imINPs (yellow arrows) when Six4 is overexpressed in type II NBs. Note that Dpn⁺ mINPs are eliminated in the lineage outlined by the dashed line. (C-C') *pros¹⁷*/+ mutant type II NB lineages have normal mINPs (white arrowheads) associated with Ase⁻ type II NBs (white arrows). (D-D') mINPs (white arrowheads) are largely rescued in all lineages when Six4 is overexpressed in *pros¹⁷*/+ heterozygous mutant type II NB lineages even though Ase is still ectopically expressed in a subset of type II NBs (open yellow arrows) and their newly generated imINPs (open yellow arrowheads). White arrows point to type II NBs without the ectopic Ase expression. (E-E") *pros¹⁷*/+ mutant type II NBs and imINPs have normal PntP1 expression in type II NBs (white arrows) and Ase⁻ (open white arrows) and Ase⁺ (yellow arrows) imINPs. Note that imINPs, particularly the Ase⁻ imINPs, usually have much higher expression of PntP1 than type II NBs. (F-F") When Six4 is overexpressed in *pros¹⁷*/+ mutants, PntP1 expression is partially still reduced in both Ase⁺ type II NBs (open yellow arrows) and Ase⁻ type II NBs (white arrows) although the expression of PntP1 in the Ase⁺ type II NBs is partially restored. The expression of PntP1 in Ase⁻ imINPs (open white arrows) and Ase⁺ imINPs (yellow arrows) is also reduced to levels that is comparable to or even lower than that in the type II NBs in the same lineages. (G-I) Quantifications of the number of type II NB lineages with mINPs (G), number of mINPs per lineage (H), and number of Ase⁻ type II NBs (I) in brains with indicated genotypes. The number on top of each bar represents the number of brains lobes examined. ***, $p < 0.001$; NS, not significant. (J-K) Quantifications of the relative PntP1 staining intensity in type II NBs (J) or Ase⁻ imINPs (K) in brains with indicated genotypes. The average staining intensity of PntP1 in the wild type type II NBs or Ase⁻ imINPs is normalized to 1. (L) Schematic diagrams show overexpressed Six4 suppresses PntP1's expression and activity, leading to ectopic Ase expression in type II NBs and nuclear expression of Pros in imINPs, resulting in the subsequent premature loss of mINPs or transformation of type II NB lineages to type I NB lineages.
(TIF)

**S6 Fig. Six4 expression is not regulated by Erm or PntP1.** (A-C') Similar Six4-GFP expression levels in type II NBs, imINPs, and mINPs are observed in wild type (A-A'), Erm knockdown (B-B'), and PntP1 knockdown (C-C') type II NB lineages. Type II NBs lineages (outlined by dashed lines) are labeled by mCD8-RFP driven by *pntP1-GAL4* and counterstained with antibodies against Dpn, Ase, and GFP. White arrows, type II NBs; yellow arrows, imINPs; white arrowheads, mINPs; yellow arrowheads, GMCs. Scale bars equal 10μm. (D) Quantifications of Six4-GFP staining intensity in type II NBs or Ase⁻ imINPs in wild type, Erm knockdown, and PntP1 knockdown type II NB lineages. NS, not significant.
(TIF)

**S7 Fig. Six4, PntP1 and Erm likely form a trimeric complex.** (A-B) Transfected Flag-PntP1 but not endogenous PntP1 are detected by western blot (A) or immunostaining (B) in S2 cells using the anti-PntP1 antibody or anti-Flag antibody. S2 cells are labeled with transfected

Act5c-Myc-EGFP and scale bars equal 10μm in (B). (C-E) Co-IP of Six4-HA and Flag-PntP1 (C), Myc-Erm and Flag-PntP1 (D), and Six4-HA and Myc-Erm (E) from S2 cells with graded concentrations (150mM– 1200mM) of NaCl. Note that interactions between these protein pairs can still be observed but are compromised at higher concentrations of salt with the interaction between Six4-HA and Myc-Erm being compromised most. (F) Co-IP of Six4-HA, Flag-PntP1, or Myc-Erm by the anti-HA antibody (lanes #1–8) or anti-Flag antibody (lanes #9–16) with 150mM or 1200mM of NaCl from S2 cells that express a combination of any two or all three of these proteins. Note that when all these three proteins are expressed together in S2 cells, they can be pulled down in the same complex (lanes #2, #6, #10, and #14). Because of unusually high background due to unknown reasons, the bands for co-IPed Six4-HA with the anti-Flag antibody were not shown, but the 2% input shows that Six4-HA was indeed expressed when its expression DNA construct was transfected into S2 cells. (F'-F") Quantifications of normalized % co-IPed Flag-PntP1 or % co-IPed Myc-Erm by the anti-HA (F') or anti-Flag (F") antibody in the indicated lanes in panel (F). The % co-IP was calculated by dividing the co-IPed protein by the 100% input of the co-IPed protein, then normalized by the IPed protein to correct the differences in immunoprecipitation. Note that the % co-IPed Myc-Erm by the anti-HA antibody with 1200mM of NaCL is significantly increased when Flag-PntP1 is co-expressed in S2 cells (compare lane #6 with lane #8). The % co-IPed Erm by the anti-Flag antibody with both 150mM and 1200 mM of NaCl is also consistently increased although not statistically significant when Six4-HA is co-expressed (compare lane #10 with lane #9, and lane #14 with lane #13). Values are mean ± SEM (N = 3). NS, not significant. *, $p < 0.05$, paired t-test. (F''') Spot graphs of normalized % co-IPed Myc-Erm of individual replicates included in plotting the bar graph in (F"). *, $p < 0.05$ when all replicates under both salt conditions are pooled together for a paired t-test.
(TIF)

**S8 Fig. Six4 knockdown alone does not result in ectopic nuclear Pros expression in newly generated imINPs.** Nuclear Pros is not detected in either Ase⁻ (open white arrows) or Ase⁺ (yellow arrows) imINPs in both wide type (A-A") or Six4 knockdown (B-B") type II NB lineages. Type II NB lineages are labeled with mCD8-GFP or mCD8-RFP driven by *pntP1-GAL4* and outlined with dashed lines. Scale bars equal 10μm.
(TIF)

**S1 Data. All the numeric data used to generate the graphs and tests for statistical analyses.**
(XLSX)

## Acknowledgments

We thank Drs. Y. N. Jan, J. B. Skeath, H. Y. Wang, and C. Desplan, the Bloomington *Drosophila* Stock Center, Vienna Drosophila Resource Center, Zurich ORFeome Project, and *Drosophila* Genomics Resource Center for antibodies, plasmids and fly stocks; Drs. F. Pignoni and A. S. Viczian for sharing research facility; N Jusic for technique support; Drs. R. T. Matthews, F. Pignoni, M. E. Zuber, X Li, X. B. Deng, and Q. X. Zhou for thoughtful discussion and comments.

## Author Contributions

**Conceptualization:** Rui Chen, Sijun Zhu.

**Data curation:** Rui Chen.

**Formal analysis:** Rui Chen.

**Funding acquisition:** Sijun Zhu.

**Investigation:** Rui Chen, Yanjun Hou.

**Methodology:** Rui Chen, Yanjun Hou.

**Project administration:** Sijun Zhu.

**Resources:** Rui Chen.

**Software:** Rui Chen.

**Supervision:** Sijun Zhu.

**Validation:** Rui Chen.

**Visualization:** Rui Chen.

**Writing – original draft:** Rui Chen.

**Writing – review & editing:** Yanjun Hou, Marisa Connell, Sijun Zhu.

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
