## [Decision Letter · Decision Letter 0]

19 Aug 2020

Dear Dr Zhu,

Thank you very much for submitting your Research Article entitled 'Homeodomain transcription factor Six4 maintains homeostasis of Drosophila larval intermediate neural progenitors' to PLOS Genetics. Your manuscript was fully evaluated at the editorial level and by three independent peer reviewers.

Although the reviewers  were very impressed by the amount of work presented and the knowledge displayed, they provide a set of extensive comments and some substantial concerns about the current manuscript that you need to address.

One specific point that was a concern by two of the reviewers were the may arrows shown in the figures that appear to detract from the clarity of these figures, which were also of poor quality in the PDF, likely due to the conversion. You will therefore have to change these figures and allow the readers to evaluate the phenotypes on their own, maybe marking one or two cells as examples. If the quality is improved, this will help significantly. You should also do your best to address all the other points noted.

Based on the reviews, we will not be able to accept this version of the manuscript, but we would be willing to review again a much-revised version. We cannot, of course, promise publication at that time.

If you decide to revise the manuscript for further consideration at PLOS Genetics, please aim to resubmit within the next 60 days, unless it will take extra time to address the concerns of the reviewers, in which case we would appreciate an expected resubmission date by email to plosgenetics@plos.org.

[LINK]

We are sorry that we cannot be more positive about your manuscript at this stage. Please do not hesitate to contact us if you have any concerns or questions.

Yours sincerely,

Claude

Claude Desplan

Associate Editor

PLOS Genetics

Gregory P. Copenhaver

Editor-in-Chief

PLOS Genetics

Reviewer's Responses to Questions

**Comments to the Authors:**

Reviewer #1: A prominent mechanism of expanding the pool of neural progenitors in the fruit fly brain is the generation of intermediate progenitors by neuroblasts. In the central brain there are several neural lineages (termed type II NBs) that display such a way of proliferation. While much is known on the switch from NB to INP, the mechanism that maintain INPs in their state remain less understood.

The authors show first that Six4 RNAi in type II NB lineages (under Pnt-Gal4) increases the number of NBs. Next they show that Six4 is highly expressed in type II NBs and that expression decreases in INPs. They provide evidence that Six4 knock-down does not affect asymmetric cell division. A set of genetic experiments using specific genetic markers (e.g. Erm, PntP1, BTD and Pros for early vs late INP) further supports that the Six4 and Erm are critical for early/immature INPs by regulating PntP1- the authors provide support that this is likely a direct action.

In summary I enjoyed reading the manuscript, it’s an intriguing addition to the type II mode of neurogenesis. The data are overall well-presented and of high quality. I only have a couple comments that were unclear to me.

>I do see the mechanisms of Six4 and Erm to act within the lineage, but I am not sure how an early defect of splitting or duplicating lineages can be excluded? Among these lines in principle the additional NBs should be observed within a lineage (fig1), however it is unclear to me where individual lineages have their boundaries and where extra NBs are localized.

> the main point of Figure 2 is to show Six4 expression, however while I agree that a triple/quadruple staining is informative, the way the wildtype data is represented is a bit confusing. Some image without the linage-depiction lines should be provided.

Some expression remains in the RNAi knock-down. It would be helpful if the authors could comment on this.

Reviewer #2: The manuscript by Rui Chen and colleagues addresses a very interesting question in neural development; namely, the mechanisms by which neural stem cells regulate the ultimate number of neurons and glial cells that are produced. These cells must prevent themselves from remaining stem cells to long but at the same time must prevent them from differentiating too quickly. When I read the abstract and introduction, I was very excited about his paper. The topic is very interesting and the authors had proposed a very nice mechanism.

However, I have significant concerns about the results section. I had a very hard time matching the text of the results with the figures.

1. My most significant issue is that had the authors had not placed arrows on each panel I would have not been able to see the results that they described in the text. I personally think that phenotypes should be displayed in an unbiased manner for the reviewer to evaluate. I don’t think our attention should be drawn to any particular region of an image. This paper is pretty egregious example – each panel with the main figures are littered with arrows – Figure 6K is a good example of what I am talking about. I tried to imaging what the figures would look like without the arrows. I concluded that I would not have been able to see what the authors are describing.

2. In most of the images, the authors have placed arrows on cells that look identical to neighboring cells that do not have arrows. Thus it is difficult for me to agree with many of the conclusions in the text.

3. The figures (at least the version that I downloaded) are of pretty poor quality. In many cases, I cannot even use the arrows as a guide. It looks like they are not pointing to any particular cell type.

4. There are some quantifications that do not match the figures. For example, in figure 3 the expression of 2 copies of UAS-Six4 RNAi using erm-GAL4 is supposed to generate 1-2 additional type II NBs – however in figure 3B the authors are pointing to the same number of type II NBs as there are in the wild type shown in figure 3A. The chart in figure 3E indicates that there are 9 type II NBs on average in the mutant genotype.

5. I think images need to be presented in a completely unbiased manner. Use arrows sparingly and allow the reviewer to make up his or her own mind without guidance on the panels. You might need to provide higher magnification images as many of the cells that you are pointing too are too small to us to analyze. And you might consider altering the color combinations to make your results easier to see. The differences between magenta and red cells is sometimes not clear in the present images.

6. You might also consider generating a new figure (figure 1) in which you provide the readers with a set of schematics that describe the neuroanatomy of the brain – something akin to figure 3G. Start with a zoomed out view of the entire brain and make a series of successive schematics that progressively zoom into the area of the brain that we will be looking at in each panel.

Reviewer #3: In this manuscript Chen et al. explore the mechanisms involved in the integrity and homeostasis of intermediate neural precursors (INPs). For that, they use a multitude of genetic and biochemical tools to disclose the mechanisms regulating fate-fidelity in neural stem cells and progenitor cells, using Drosophila’s neuroblasts (NBs) and immature precursor progenitors (INPs) as their model. In order to identify genes that are necessary for proper INP development, the authors conducted a screen where they specifically abolish the levels of transcription factors within type II NBs and then described how it affected proper lineage formation, in terms of number of NB, immature INPs (imINPs) and mature INPs (mINPs) within the central brain lobes. From this screen, they identified that downregulation of Six4 (that is consistently and more strongly expressed in type II NB, with progressive but consistent expression throughout the lineage) resulted in supernumerary NBs (from 8 to an average of 16 NBs per brain lobe), without affecting the number of mINPs. Also, they properly show that overexpression of Six4 in a RNAi background successfully rescues the phenotype. Thus, the authors conclude that Six4 is required to prevent supernumerary type II NBs in the larval brain. Then, it is excluded that supernumerary NBs arise from defective asymmetric division, while induction of Six4 RNAi expression in imINPs through an earmuffGal4 driver results in a mild but consistent increase in the number of NBs present (from 8 to 9-10). After, they show that abrogation of Six4 within NBs results in similar levels of PntP1 within NBs and imINPs, but that expression persisted in mINPs, indicating delayed suppression of PntP1. Not only that, but persistent levels of PntP1 promoted by Six4 depletion does not affect levels of Erm. Altogether, this indicates that both Six4 and Erm are involved in PntP1 repression in INPs. In line with the previous finding, overexpression of Six4 within NBs results in type II to type I transformation in 25% of the cases and Pntp1 decreased expression. Also, type II that retained their identity generated fewer INPs. Furthermore, the authors show that, like Erm, Six4 is capable of suppressing PntP1 expression and antagonize the activity of the latter. Hence, by ensuring proper inhibition of PntP1, Six4 is capable of granting imINP to become fate committed. To further elucidate the regulatory role of Six4, the authors show that (i) Six4 is capable of functionally replace Erm to prevent imINP dedifferentiation; (ii) Six4 and Erm independently co-immunoprecipitate with PntP1; (iii) Six4 and Erm coimmunoprecipitate with each other, indicating that these proteins interact with each other. While authors show that Six4 is capable of inhibiting PntP1 during INP maturation, the role of Six4 in NBs is still unclear. While overexpression of Six4 in NBs is capable of inhibiting PntP1, what is its physiological role in NBs? For that, authors co-inhibit PntP1 and Six4 in NBs in hopes that by silencing Six4 it would be possible to restore levels of PntP1 and rescue of the phenotype observed in single PntP1 knockdown (supernumerary Asense-positive NBs and mINP depletion). Nevertheless, the phenotype remained the same in terms of Ase-positive NBs. On the other hand, mINPs were completely depleted in this scenario and the supernumerary phenotype of type II NBs was also inhibited. Nevertheless, the authors show that, at a physiological level, Six4 may in part contribute to the suppression of prospero in newly-generated imINP, such as PntP1 does.

Overall, the authors show great knowledge in terms of the field and the literature available for the full comprehension of this topic. Not only that, but it is possible to observe that the experiments have been well thought out, well-executed, and with a strong rationale to support them. Moreover, the manuscript is well written and can be understood by someone in the field. Due to the nature of the work, the manuscript can be a bit more difficult to follow by a non-specialist. Altogether, these novel results increment the knowledge we have today regarding NBs and INPs homeostasis and contribute greatly to the field.

Point-by-point comments:

(related to Figure 1)

It would be very interesting to also include the cell-type counts per lineage, so the reader has the understanding on whether there are supernumerary NBs within one lineage or if the supernumerary NBs form their independent lineages (from the image, it seems that is the second case). This would corroborate that supernumerary NBs do not form by incorrect asymmetric division, but it would also not be evident that they would form by INP fate-reversion;

(related to Figure 2)

Possibly, the image could be simplified: maybe just keep the arrows/arrow heads in one of the lineages and/or just keep them in the image with the merged channels;

(line 157)

Some CB type I NBs and respective newly-formed GMC and GMCs from the VNC (but not NB) express Six4. This is very interesting in terms of neural stem cell heterogeneity and eventual adult neuronal diversity. Are always the type I NBs and/or GMCs spatially located in the same regions of the brain?;

(related to Figure 3)

While the experiments are consistently well designed throughout the manuscript, it is unclear what type of control it is used. For instance, in this case (mostly because the phenotype is so subtle), it would be important that the control animals also express two copies of a non-targeting RNAi (like a UAS-mcherry RNAi). Also, in this case, it would be important to know the number of NBs per lineage to understand where the extra NBs formed are located. This would be interesting, considering that the phenotype might arise early in development, during NBs specification (resulting in more NBs and more lineages), or if later during larval stages (resulting in more NBs within one single lineage, which in the representative Figure seems to be the case). It would also be interesting to further dissect the mechanisms of fate reversion by the absence of Six4 through the use of reporter lines (targeting fate-associated proteins) and long-term live cell and/or whole-brain imaging. Figure 3A is missing the scale bar;

(related to Figure 4)

For simplification, reduce the number of arrows (maybe just keep them in some of the panels);

(related to Figure 5)

For simplification, reduce the number of arrows (maybe just keep them in some of the panels);

(line 334)

While it is a strong possibility that Six4 binds directly to Erm, these experiments do not necessarily allow us to conclude that. Instead, it is possible to conclude that both are in close proximity/interact within the same complex/interact within the same chromatin context. Also, related to Figure 6P, it would be interesting to see whether you would see PntP1 being coIPed with Six4-HA or Myc-Erm, since none of the coIP experiments show that the three proteins are coIPed together: the upper panel shows that when Six4-HA is IPed it brings down Myc-Erm with it and, similarly, when Myc-Erm is IPed, it brings down Six4-HA. In addition, to better understand these questions, it would interesting to test the coIP using higher salt concentrations, to have a better idea how strong of an interaction it is. Also, it would be interesting to further address this question (not necessarily for the scope of this specific paper), like doing chromatinIP or targeted-DamID to address the chromatin binding of Six4/Erm/PntP1, to further dissect this regulatory axis at the molecular level;

(line 334-336)

The authors suggest that inhibition of PntP1 in imINPs putatively occurs via physical interaction with Six4 and Erm. Because this would take place after protein translation, as a post-translational mechanism, it does not justify how transcript levels of PntP1 drastically decrease over lineage progression (unless there is a positive feedback loop where PntP1 regulates its own expression);

(related to Figure 6)

Just like previously suggested, for simplification purposes arrows can be removed;

(line 418-424)

Would it be possible to use an erm-Gal4 in the II chromosome in combination with erm-Gal4 on the III chromosome? Both are expressed in imINPs, but in slightly different stages (either in Ase-negative – R9D11-Gal4 (II) – or in Ase-positive imINPs – R9D11-Gal4 (III)). Maybe this strategy would enlarge the timeframe of expression of the RNAi. Also, the second hypothesis could be easily addressed by using the modERN line expressing Six4-GFP, to see whether induction of the RNAi against Six4 with erm-Gal4 is not enough to decrease levels of Six4 in INPs;

(line 463 onwards)

Authors discuss whether the presence of Erm and Six4 is required for complex stability. First, while results are very convincing and well curated, I think it is too premature to state that they form a proper complex. It seems too strong of a statement given the results available. From their IP studies, when they express Six4-HA or Myc-Erm, the interaction with PntP1 is of the same extent (also Erm and Six4 are not endogenously expressed in S2 cells, so they cannot compensate each other). By performing a salt gradient, maybe it would be possible to study the stability of this complex. Also, it has been shown that transient overexpression of Insensible in a brat null induces supernumerary type II NBs to synchronously transition into INPs, thus leading to an enrichment of INPs within the brain (Rives-Quinto et al., bioRvix, doi: https://doi.org/10.1101/2020.02.03.931972). This could be a good strategy to do this same experiment in a Drosophila brain context;

(Related to Supp Figure 1)

There is a typo (ventral instead of vental).

**Have all data underlying the figures and results presented in the manuscript been provided?**

Reviewer #1: Yes

Reviewer #2: Yes

Reviewer #3: Yes

PLOS authors have the option to publish the peer review history of their article (what does this mean?). If published, this will include your full peer review and any attached files.

Reviewer #1: **Yes: **Simon Sprecher

Reviewer #2: No

Reviewer #3: No

---

## [Decision Letter · Decision Letter 1]

7 Dec 2020

Dear Sijun,

Thank you very much for submitting your revised version of the research article entitled 'Homeodomain transcription factor Six4 maintains homeostasis of Drosophila larval intermediate neural progenitors' to PLOS Genetics.

The manuscript was fully evaluated at the editorial level and by two of the initial independent peer reviewers. One reviewer who was positive was not consulted officially.

Although the reviewers agree that the paper is potentially of significant interest, and appreciated your efforts to address the initial set of comments, both of them still have very strong objections to the way you interpreted the data, and they suggest extensive revisions.

It is rare when we allow a second major revision but we are going to give you a chance to address these comments. We do not ask you to do new experiments but we must rewrite the paper (and change the title) in order to much better represent the nature of the conclusions that can be drawn from the data. The reviewers have provided extensive guidance as to what to do.

Sp, please listen carefully to the recommendations and resubmit a revised paper that will be sent back to reviewers, who, I hope, will be convince this time

If you decide to revise the manuscript for further consideration at PLOS Genetics, please aim to resubmit within the next 60 days, unless it will take extra time to address the concerns of the reviewers, in which case we would appreciate an expected resubmission date by email to plosgenetics@plos.org.

[LINK]

We are sorry that we cannot be more positive about your manuscript at this stage. Please do not hesitate to contact us if you have any concerns or questions.

Yours sincerely,

Claude Desplan

Associate Editor

PLOS Genetics

Gregory P. Copenhaver

Editor-in-Chief

PLOS Genetics

Reviewer's Responses to Questions

**Comments to the Authors:**

Reviewer #2: I appreciate the authors efforts to address the comments of myself and the other two reviewers. The additional schematic drawings are very helpful. I also appreciate the authors taking the suggestion of two of us and removing many of the arrows and arrowheads from various figures. As I mentioned in my first review, I find the topic of this paper very interesting and it is addressing a very important question – namely how are intermediate neural progenitors prevented from either dedifferentiating or differentiating too quickly. As the authors point out, maintaining the balance of INPs is critical for producing a brain of the right size and cell numbers.

I read the paper several times and I still am having trouble seeing the phenotype just from the images. I have downloaded the original files and while they are of very high resolution it is really difficult for me to see the difference among the different genotypes. In my initial review I had requested that the authors remove the arrows and arrowheads since I wanted to see if I could look at the brain images and see if I came to the same conclusion as the authors. Without the graphs documenting the quantifications, I would not be able to see the difference between many of the panels. For a non-expert it is not clear why some cells are counted while others are not. For example, in panels 2B,C,D the authors place arrows on certain cells. But there are other cells that show the same expression profile that are not marked with arrows. This is true of many of the panels throughout the paper. I think this is going to be problematic for the non-expert and will reduce the enthusiasm for the paper.

I think the quantifications that are presented throughout the paper suggest that Six4 is playing an essential role in this maintaining INP balance in numbers. It is just very hard for me to see it without the arrows/arrowheads and charts. I guess I am struggling with understanding how the authors selected some cells and not others even though both cells express the same set of markers.

I also feel that some of the images are very small. This was a problem with the first version. Even looking at the high resolution images does not help. It is still very hard to see what is being pointed to in figure 7A,B,F,G,H.

The authors mention that Six4 is functioning after larval hatching. Using a tub-GAL80ts, the authors allowed for expression of the RNAi line only after larval hatching and see that each brain lobe had one extra type II NB. This is in contrast to flies that don’t have the tub-GAL80ts, which results in 16 NBs per brain lobe. The authors conclude that since they see one extra NB it must mean that Six4 is functioning after larval hatching. I would think that since most of the extra NBs are not generated when the RNAi line is expressed after larval hatching it would mean that Six4 is actually functioning during embryonic stages.

The manuscript would also be helped if the authors could include a molecular model of how NB progress to imINPs to mINPs to GMCs. There is a lot going on in this pathway and the authors do an excellent job of describing it within the introduction. A diagram would really help the reader.

Reviewer #3: First of all, I appreciate the efforts the authors made to address all the comments, questions, and suggestions proposed by all the reviewers. Overall, while the manuscript was very well written in its initial version, in some instances the figures were a bit difficult to interpret, thus making difficult the understanding of the results. By simplifying and clarifying their meaning, it was clearer what the authors report in the text. Also, in their point-by-point answers to the reviewers, the authors were objective and up-front, addressing most of the questions appropriately. In addition, as suggested by another reviewer, the authors included a much-appreciated schematization of the larval brain, incredibly useful for those that are non-specialists.

I have a number of concerns regarding some of the pivotal findings (some of them serving as the main premise of the paper). For instance, and related to the results depicted in Figure 1, one of the things that was not clear in the initial version of the manuscript, was whether the supernumerary neuroblasts arising upon Six4 depletion were: (i) neuroblast forming independent lineages and those lineages were virtually identical to a wildtype one (thus aligning with the idea that the phenotype promoted by loss of Six4 arises during early NB specification) or; (ii) the ectopic neuroblasts would co-exist withing the same lineage thus supporting the idea of defective asymmetric division (which the authors do not observe) or dedifferentiation/fate reversion of INPs. According to the answer provided by the authors “4% of Six4 knockdown type II NB lineages had multiple type II NBs, while the other 96% of NBs formed independent lineages (…)”. First, if the phenotype is promoted by dedifferentiation of INPs, one would expect lineages with ectopic NBs, that is only verified in 4% of the cases (considering that each lobe has 8 lineages, this means the phenotype is observed in one lineage every 4 lobes or 2 brains). Also, if the NBs are formed at the expense of INPs, one would expect decreased numbers of this cell type (and GMCs) along the affected lineages, which is not really the case (as shown in Figure 1J).

Another concern regards the claims that the phenotype takes place after larval hatching, thus excluding that NBs specification during embryonic stages would not be affected. In fact, this was one of the interrogations that I and other reviewers had when looking at the figures provided during the initial submission of the paper. To tackle this problem, the authors controlled the expression of the Six4 RNAi with tubGal80ts, initially rearing the animals at the restrictive temperature and only shifting them to the permissive temperature after larval hatching. In this context, as shown in Figure S1, the authors observed the appearance of one more NB per lobe (in average) in opposition to the 16 NBs they observe when animals without expression of tugGal80ts. Because the phenotype is completely distinct and much stronger in one case versus the other (and while one can argue whether the RNAi is efficient during larval stages), these results indicate that the Six4 is actually exerting its activity during embryonic stages. Having that, and while it is not excluded that Six4 exerts a regulatory role during larva stages, it should not be excluded that the phenotype (supernumerary neuroblasts caused by NB duplication or unfaithful specification during embryonic stages).

Later, to investigate whether Six4 generates supernumerary NBs through dedifferentiation of INPs, the authors decide to abolish Six4 levels within INPs and address the number of NBs. Again, as previously, the phenotype is extremely discreet (only 1 to 2 more NBs per lobe, with only 1 type II lineage having more than 1 NB). The phenotype was slightly stronger when Six4 RNAi was performed in erm heterozygous background (Figure 3 E), but still very variable (is most of the cases – 5 in a total of 8 – the phenotype is the same in erm heterozygous background alone, Figure 3D and G). Not only that, but from what is possible to understand from Figure S4 D, E, levels of Six4 within INPs are virtually the same as the control, indicating that the RNAi is not effective when its expression is driven by ermGal4 (in opposition to pointedGal4).

At this point, I would not advise the execution of complementary experiments. Instead, I think that the text can be reorganized/re-written in order to better reflect the results in a completely unbiased way. Taking that into consideration, I believe that the manuscript is of interest to the field and I recommend its publication if the authors address the critical concerns described previously and described one-by-one bellow:

- I would recommend a change in the title, in order to reflect better the results shown in the manuscript.

- From line 128 to 130, the authors claim that “Co-existence of multiple type II NBs in single lineages indicates that these extra type II NBs could be just generated late during larval development and did not have enough time to establish their own independent lineages yet.”. Wouldn’t you expect that, after each round of NB division, the newly formed INP (that usually takes up to 6h to maturate and divide) would fail to commit to its fate and revert back to a NB-like state? If so, shouldn’t there be even more NBs per lobe and per lineage? And why would this occur on such a relatively small scale? Considering that you have 16 NBIIs per lobe upon Six4 depletion, it means that, on average, only one 1 INP per lineage dedifferentiated? Why that only one INP would be affected? Are these newly formed NBs any different from the primary ones? These questions are in the context of INP dedifferentiation upon knockdown of Six4.

- From line 154 to 156 the authors conclude that knockdown of Six4 during larval stages (using tubgal80ts) supports that the supernumerary neuroblasts arise during larval development and not during NB specification. Unfortunately, these results do not necessarily support that statement. In no place in the paper should be excluded the embryonic contribution of Six4 (in fact, Six4 could play two distinct roles during animal development: (i) initially through regulation of NB specification and (ii) later in the maintenance of INP pool). This should be clearly stated in the results section and, if appropriate, the authors could develop on this in the discussion section.

- From line 204 to 205 the authors conclude that “Although the phenotype is weak, the consistent generation of extra type II NBs and the existence of multiple type II NBs in single lineages indicate that the supernumerary type II NB phenotype likely results from dedifferentiation of imINPs.”. This is clearly an overstatement and the sentence/section should be changed in order to factually report what is observed. From the image provided, it is not clear the co-existence of 2 or more NBIIs per lineage and, indeed, the phenotype is very weak (likely due to the inefficiency of the RNAi in INPs). At this point, and without any further experiments, it is difficult to say that, by itself, abrogation of Six4 leads to dedifferentiation of INPs (you would require further experimentation to appropriately test this).

- From line 207 to 210. I do think that this result is more indicative (but not conclusive) regarding the role of Six4 in INP dedifferentiation. Since the idea is that ectopic NBs are formed at the expense of INPs, it would be interesting to determine the number of mINP in this specific experiment (considering that you have staining for both Ase and Dpn, this should be fairly simple to do).

- When pntp1 and erm start to be introduced in the results section (namely regarding synergistic role with Six4), it would be interesting to include a scheme depicting the molecular pathway through which a type II lineage progresses (NB to imINP, the maturation process of the INP and its asymmetric division to give rise to GMCs).

**Have all data underlying the figures and results presented in the manuscript been provided?**

Reviewer #2: Yes

Reviewer #3: Yes

PLOS authors have the option to publish the peer review history of their article (what does this mean?). If published, this will include your full peer review and any attached files.

Reviewer #2: No

Reviewer #3: No

---

## [Decision Letter · Decision Letter 2]

15 Jan 2021

Dear Dr Zhu,

We are pleased to inform you that your manuscript entitled "Homeodomain protein Six4 prevents the generation of supernumerary Drosophila type II neuroblasts and premature differentiation of intermediate neural progenitors" has been editorially accepted for publication in PLOS Genetics. Congratulations! It took some time to reach this point but I am sure that you are yourself happy about the content of the new manuscript. The  reviewers were impressed by the way you handled the comments, critiques and suggestions, and I am happy about the outcome.

Yours sincerely,

Claude Desplan

Associate Editor

PLOS Genetics

Gregory P. Copenhaver

Editor-in-Chief

PLOS Genetics

Comments from the reviewers (if applicable):

Reviewer's Responses to Questions

**Comments to the Authors:**

Reviewer #2: The manuscript is much improved and I am now in favor of its publication.

Reviewer #3: As before, I would like to appreciate the efforts made by the authors to address all the questions/concerns raised by the reviewers. In its last version, the manuscript is well written and easier to comprehend. Also, and upon analysis of the answers provided by the authors (and respective alterations done in the manuscript itself), the authors were very pragmatic and comprehensive, allowing us to better understand the data and their point of view. Overall, I believe that the manuscript significantly increments the knowledge of the field and, for that reason, I believe it meets the publication standards required by the journal.

**Have all data underlying the figures and results presented in the manuscript been provided?**

Reviewer #2: Yes

Reviewer #3: **No: **

PLOS authors have the option to publish the peer review history of their article (what does this mean?). If published, this will include your full peer review and any attached files.

Reviewer #2: No

Reviewer #3: No

**Data Deposition**

http://datadryad.org/submit?journalID=pgenetics&manu=PGENETICS-D-20-01043R2

**Press Queries**

---

## [Editor Report · Acceptance letter]

2 Feb 2021

PGENETICS-D-20-01043R2 

Homeodomain protein Six4 prevents the generation of supernumerary *Drosophila* type II neuroblasts and premature differentiation of intermediate neural progenitors 

Dear Dr Zhu, 

We are pleased to inform you that your manuscript entitled "Homeodomain protein Six4 prevents the generation of supernumerary *Drosophila* type II neuroblasts and premature differentiation of intermediate neural progenitors " has been formally accepted for publication in PLOS Genetics! Your manuscript is now with our production department and you will be notified of the publication date in due course.

With kind regards,

Alice Ellingham

PLOS Genetics

On behalf of:
